
# PALEOSTRIPv1.0 - a user-friendly 3D backtracking software to reconstruct paleo-bathymetries

Florence Colleoni[1], Laura De Santis[1], Enrico Pochini[1], Edy Forlin[1], Riccardo Geletti[1],
Giuseppe Brancatelli[1], Magdala Tesauro[2,3], Martina Busetti[1], and Carla Braitenberg[2]

[1]National Institute for Oceanography and Applied Geophysics - OGS. Borgo Grotta Gigante 42/c, 34010 Sgonico (TS), Italy
[2]University of Trieste, Department of Mathematics and Geoscience, Trieste, Italy
[3]University of Utrecht, Department of Geosciences, the Netherlands

**Correspondence:** Florence Colleoni (fcolleoni@inogs.it)

**Abstract.** We present PALEOSTRIPv1.0, a MATLAB open-source software designed to perform 1D, 2D and 3D backtracking of paleo-bathymetries. PALEOSTRIP comes with a Graphical User Interface (GUI) to facilitate computation of sensitivity tests and to allow the users to switch on and off all the different processes and thus separate the various aspects of backtracking. As such, all physical parameters can be modified from the GUI. It includes 3D flexural isostasy, 1D thermal subsidence and possibilities to correct for prescribed sea level and dynamical topography changes. In the following we detail the physics embedded within PALEOSTRIP and we show a few applications on a drilling site (1D), a transect (2D) and a map (3D), taking the Ross Sea (Antarctica) as a case study. PALEOSTRIP has been designed to be modular and to allow user to insert their own implementations.

## 1 Introduction

On-going climate changes are urging the scientific community to project climate future evolution in response to carbon emissions trajectories (e.g. the Shared Socio-economical Pathways Riahi et al., 2017). The Coupled Model Intercomparison Project (CMIP), now ending phase 6 (Eyring et al., 2016), has been producing a large amount of climate projections until 2100. However some of the climatic variables, for example, the deep ocean, the carbon cycle and the ice sheets and glaciers, react more slowly to climate changes (e.g. Colleoni et al., 2018b; Noble et al., 2020), though showing evidence of changes over the past decades already (Caesar et al., 2021). Their main response has yet to be observed and is likely to happen beyond the 21st century, which is encouraging climatologists to project changes on longer timescales of few centuries to millennia (e.g. IPCC, 2013, 2019). Projecting to such timescales implies to design corresponding, realistic carbon emission trajectories and so far, millennial scale emissions trajectories are just extension of existing emission scenarios beyond 2100 (e.g. Golledge et al., 2015; DeConto and Pollard, 2016). At this point, the exercise becomes difficult and this is when reconstructing past climates becomes important. It allows to test the response of the Earth's climate under different, but realistic, atmospheric greenhouse



gas concentrations (GHGs) (e.g. Haywood et al., 2019). GHGs higher than present-day or future ones, i.e. larger than 400 ppm for atmospheric $CO_2$, can only be found for times older than 3 million years ago (e.g. Zhang et al., 2013; Bracegirdle et al., 2019). Going back to those times, or even before, it is likely that the tectonic setting responsible for other boundary conditions,

such as oceanic gateways, elevation of mountain ranges, continental margin expansion, as well as the location and extent of continental masses themselves, differed from that of the present-day. Nevertheless, simulating and reconstructing past climatic conditions can bring useful hints on how future climate might evolve and also help narrowing the range of likely long-term carbon emissions trajectories.

Reconstructing past topographies and bathymetries is fundamental for paleoclimate and paleo ice sheet simulations (e.g.

Otto-Bliesner et al., 2017; Colleoni et al., 2018a; Straume et al., 2020; Paxman et al., 2020). The numerous oceanic deep drilling campaigns that occurred over the past decades have the potential to constrain such reconstructions, but not when part of the information has been overprinted during geological time by other tectonic or climatic processes. In addition, during sediment deposition, the bathymetry itself changes due to different processes, such as the loading of accumulated sediments, thermal subsidence acting on extended continental crust, subsidence/uplift induced by mantle dynamics (dynamic topography)

or sea level changes (e.g. Kirschner et al., 2010; Celerier, 1988; Sclater and Christie, 1980). Other external factors also influence the bathymetry, as for example ice loading in polar areas. By accounting for all those factors and processes, by decompacting and removing overlying sediments, it is possible to backtrack the bathymetry, and the paleo-water depth, of a chosen specific time interval. Conversely, if the target of the study is to reconstruct the burial history of a sedimentary basin, the technique is the same, but needs constraints on the paleo-water depths and is called "Backstripping" (Steckler and Watts, 1978).

There are a few existing open-source backstripping or backtracking codes. Flex-Decomp by Badley Geoscience Ltd. (Kusznir et al., 1995) allows to perform 2D-flexural backstripping or backtracking and its code is not open-source. A 3D version of Flex-Decomp exists but is not open-source (Roberts et al., 2003) and is used exclusively by Badley Geoscience Ltd., and not available externally. It comes along with the sister program Stretch by Badley Geoscience Ltd., a software to compute forward modeling of basin evolution that provides spatially variable stretching factors cross-section to be used by Flex-Decomp. BasinVis (Lee

et al., 2016, 2020) is a MATLAB open-source code with a graphical user interface. It calculates compaction trend from input drill sites (1D) lithological units. The estimated compaction trends can be applied in thickness restoration of stratigraphic units and subsidence analysis data that can be spatially interpolated between input drill sites to reconstruct a temporal basin evolution. PyBacktrack, is a 1D-backtracking and backstripping open-source code (Müller et al., 2018b) aimed at reconstructing paleo-bathymetries. It allows to process drilling sites both on oceanic and continental crust and can be connected to the suite

of Geodynamical open-source software GPlates (https://www.gplates.org/) and benefit from geodynamical corrections related to cinematic, tectonic and geodynamic models of plates tectonic movements through time. DeCompactionTool (Hölzel et al., 2008) propose a similar approach than pybacktrack, for 1D well, but allows to perform a Monte-Carlo like analysis, i.e., perform a large number of 1D runs based on a possible range of main physical parameters defined by admissible minimum and maximum values, to provide a quantitative estimate on the backstripping error.

3D flexural backstripping and backtracking are needed to reconstruct basin-wide or continental-wide areas to force climate and ice sheet models. A few studies mentioned the use of 3D backstripping (e.g. Scheck et al., 2003; Hansen et al., 2007;





Smallwood, 2009; Amadori et al., 2017), and a very limited number provides with 3D flexural backstripping (Roberts et al., 2003; Steinberg et al., 2018; Roberts et al., 2019).

Here we present PALEOSTRIP, a MATLAB open-source software designed to perform 1D, 2D and 3D backtracking of paleobathymetries. PALEOSTRIP comes with a Graphical User Interface (GUI) to facilitate computation of sensitivity tests and allows the users to switch all the different processes on and off and thus separate the various aspects of backtracking. As such, all physical parameters can be modified from the GUI. It includes 3D flexural isostasy, 1D thermal subsidence and possibilities to correct for prescribed sea level and dynamical topography changes. In the following we detail the physics embedded within PALEOSTRIP and show a few applications on a drilling site (1D), a transect (2D) and a map (3D), taking 65 the Ross Sea (Antarctica) as a case study.

## 2 Model Framework and requirements

PALEOSTRIP is a MATLAB open-source code developed under the GNU General Public License v3.0. It is composed of a set of routines called by a graphical interface from which users can load data, change physical parameters, plot and save results (Fig. 1). The code is distributed on Github (https://github.com/flocolleoni/PALEOSTRIPv1.0) along with a User Manual 70 providing all necessary explanations and examples on how to input data and use PALEOSTRIP functionalities.

PALEOSTRIP has been designed and coded with MATLAB (R2019b) and run on any operative system. Most of the code should be compatible with previous versions of MATLAB, provided that it includes the App Designer toolbox. The code is incompatible with GUIDE and from MATLAB R2020, GUIDE is not available anymore. PALEOSTRIP cannot be run without its graphical interface, thus a version of MATLAB with the App Designer toolbox is necessary.

### 2.1 PALEOSTRIP Graphical User Interface

PALEOSTRIP Graphical User interface (GUI) is composed of two different tabs. The first one is dedicated to input data, physical parameters and choices of physical methods for each of the processes accounted for in backtracking (Fig. 1). The second one is dedicated to plotting backtracked data and saving results (see Sect. 6). The GUI can be launched either through the MATLAB main interface and by double clicking on the corresponding App Designer GUI file, or by exporting it as a stand-80 alone application that can be executed without opening MATLAB. We do not provide a built-in PALEOSTRIP application, since the compilation of the applications depends on the Operative System on which it is created, but we provide PALEOSTRIP code instead. As such, the user is free to export it from the App Designer tool.

### 2.2 Format of input and output data

#### 2.2.1 Coordinates system

Cartesian coordinates (in meters) are required to perform flexural isostasy calculations, as the flexural response depends on the distance from the load. Consequently, input data must be provided on a cartesian coordinates grid. PALEOSTRIP does





not run on geographical coordinates and does not provide any tool to convert input data from geographical to cartesian coordinates. However, many open-source softwares or codes exist to perform this step, such as the MATLAB Mapping Toolbox (https://it.mathworks.com/products/mapping.html), The Generic Mapping Tools 6 (https://www.generic-mapping-tools.org/), OBLIMAP2 (Reerink et al., 2016) or any other GIS software.

### 2.2.2 Input files

Horizon depths have to be provided individually in single ASCII files, defining the given quantity along with its coordinates. For example: horizon depth Z for a 1D drill site; extension along the transect and depth (X, Z) for 2D transects; the horizontal position (X,Y) and depth (Z) for 3D horizon maps. Output data are saved with the same format. PALEOSTRIP does not read and write grids in the NetCDF format, this functionality will be the implemented in a future release. Lithological parameters must be provided in a separate ASCII file and are spatially uniform.

## 3 PALEOSTRIP: backtracking

During its evolution, a submarine continental margin can experience various processes that modify its morphology. For example, a rifted basin can form in response to plate tectonics displacement that shapes the basic structure of the margin. Surface erosion from the hinterland can supply the margin with sediments filling morphological depressions if the accommodation space is large enough. The accommodation space depends on initial conditions, on the tectonic and on the thermal subsidence of the continental margin, i.e., the lithosphere and asthenosphere cooling during and after the rifting leading to a deepening of the margin through time. Eustatic and regional sea level changes modulate the available accommodation space and the distance from the sediment sources. To reconstruct the past subsidence history of a continental margin at a given time, all those processes have to be accounted for and corrected in a procedure called "Backstripping" (Steckler and Watts, 1978). Backstripping consists in decompacting and removing sediment layers iteratively back in time to reconstruct the past tectonic subsidence history of a basin or a margin (Fig. 2):

$$Z(i) = \frac{\rho_m - \rho_s}{\rho_m - \rho_w} S(i) + WD(i) - \frac{\rho_m}{\rho_m - \rho_w} \Delta SL(i) \tag{1}$$

where the first term of the right-hand side corresponds to the isostatic compensation ($\rho_s$ sediment density, $\rho_m$ mantle density, $\rho_w$ seawater density) of the $ith$ sediment layer thickness $S(i)$ accumulated on the basement, $WD(i)$ is the paleo-water depth of the $ith$ layer and the last term of the equation corresponds to the water-load correction due to sea level variations at time of the $ith$ layer. This equation solves the time-evolution of total subsidence $Z(i)$ of the basement and $WD$ needs to be provided for each $i$ layer to solve the equation. Conversely, in case the focus is on reconstructing time-varying paleo-water depths $WD(i)$, the procedure is called "Backtracking" and a total subsidence $Z(i)$ time-evolution needs to be provided as input to the equation:

$$WD(i) = Z(i) - \frac{\rho_m - \rho_s}{\rho_m - \rho_w} S(i) + \frac{\rho_m}{\rho_m + \rho_w} \Delta SL(i) \tag{2}$$





PALEOSTRIP is a backtracking software and is designed to reconstruct paleo-water depths given a provided subsidence history.

Equations above are the original backstripping and backtracking equations developed by Steckler and Watts (1978) and explained therein in major detail. Over the past decades, it has also been found that mantle convection generates changes in the
regional topography, the so-called "dynamic topography" (Müller et al., 2018a). Paleo-water depth needs to be corrected for dynamic topography changes $\Delta DynT(i)$ causing uplift or subsidence of the topography and bathymetry:

$$WD(i) = Z(i) - \frac{\rho_m - \rho_s}{\rho_m - \rho_w} S(i) + \frac{\rho_m}{\rho_m + \rho_w} \Delta SL(i) + \Delta DynT(i) \tag{3}$$

In this equation, the second and third terms in the left-hand side have been written accounting for Airy local isostatic compensation. However PALEOSTRIP also makes use of 2D and 3D flexural isostasy (See Sect. 3.2). In this case, the second and
third terms in the left-hand side of the equation represent the depth correction due to flexural response to unloading of sediment during backtracking and to water loading/unloading due to sea level changes.

In the following, we explain how each term of Eq. (3) is treated within PALEOSTRIP. Most of the equations reported below are taken from Allen and Allen (2013) if not otherwise specified. The various aspects of the backtraking procedure are explain following PALEOSTRIP workflow (Fig. 3).

## 3.1 Decompaction


The total sediment thickness $S$ accumulated at a given time can be obtained by reconstructing its compaction history. Eroded sediments that are transported to the margin deposit and accumulate wherever the accommodation space allows for it. Accumulated sediments compact through time under loading, for example, by overlying sediments. Therefore, to reconstruct the paleo-architecture of a continental margin at a given time, the sediment layers of a margin are stripped off sequentially, and
the remaining underlying sediments need to be gradually decompacted. Decompaction is thus central to backstripping and backtracking.

Compaction or decompaction of sediments both imply a change in total volume $V_{total}$ of deposited sediments mostly caused by changes in the porosity of sediments and to a lesser extent by sediment compression. In submarine environments sediments are saturated by water and their compaction implies the decrease in pore fluid pressure by expelling the water out of
the sediment layers. Conversely, decompaction involves an increase in the pore fluid pressure by injecting water within the compacted sediments. The decompaction process consists in calculating the changes in porosity of the various sediment layers to determine the amount of water that was contained in the sediments at the time of their deposition, based on their lithology. The depth of the decompacted sediment layers is recalculated on the basis of those porosity changes. Laboratory experiments have determined that for large depths, the evolution of porosity can be described by an exponential relationship rather than by
a linear empirical equation:

$$\phi = \phi_0 e^{-cz} \tag{4}$$

where $\phi_0$ corresponds to deposition porosity at the seafloor (or at surface if emerged), $c$ is the exponential slope decay coefficient (also referred as compaction coefficient herein) depending on the lithology expressed in $1/km$ and $z$ is depth in km. This empirical equation implies that $\phi_0$ is decreased by the factor of $1/e$ at the depth of $1/c$ km.



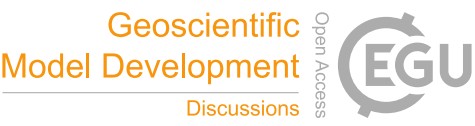

In general, the mass of the total sediment column at a given time does not change: in submarine environment the water that
is expelled from the sediments is implicitly added to the water column above the underlying sediments during the compaction
processes. Thus, only the volume of the sediment layers $V_{total}$ changes due to volume water changes $V_w$, which allows to
integrate the porosity equation (4). Considering a unit cross-sectional area, the thickness of water $W_T$ to be added to the
compacted sediment volume, in order to decompact it is given by:

$$W_T = \int_{z_i}^{z_{i+1}} \phi_0 e^{-cz} dz \tag{5}$$

which on integration gives:

$$W_T(i) = \frac{\phi_0}{c}\{e^{-cz'_i} - e^{-cz'_{i+1}}\} \tag{6}$$

where $z'_i$ and $z'_{i+1}$ are the newly decompacted depths of $z_i$ and $z_{i+1}$. Because the volume of sediment grains $V_{sed}$ remains
unchanged, the integrated compacted sediment thickness $S_{dt}$ between two given depths, considering a unit cross-sectional
area, can be written as:

$$S_{dt} = z_{i+1} - z_i - \frac{\phi_0}{c}\{e^{-cz_i} - e^{-cz_{i+1}}\} \tag{7}$$

Based on the fact that $V_{total} = V_{sed} + V_w$, the decompacted depths $z'(i)$ for a unit cross-sectional area can be inferred and the
final decompaction equation is given by:

$$z'_{i+1} - z'_i = z_{i+1} - z_i - \frac{\phi_0}{c}\{e^{-cz_i} - e^{-cz_{i+1}}\} + \frac{\phi_0}{c}\{e^{-cz'_i} - e^{-cz'_{i+1}}\} \tag{8}$$

PALEOSTRIP solves this equation iteratively (starting with $z'_i = 0$ and then for the next iterations) until convergence is reached.
Lithological parameters, i.e. the decompaction coefficient $c$ and the depositional surface porosity $\phi_0$ are prescribed in the user-
provided parameter files (See Sect. 6).

### 3.2    Isostatic correction

Sediments load the basement of the continental margin and cause a local and regional subsidence during accumulation. Water
also loads the basement due to eustatic sea level variations, leading to changes in the water depth $WD$, also caused by ocean
bottom subsidence and/or uplift through time. In addition, during the decompaction, the newly computed decompacted depths
of the remaining sediment layers need to be adjusted to account for the changes of their porosity and thus their density due to
the presence of water filling the pores. For each sediment layer, the porosity $\rho_{ds}(i)$ of the decompacted thickness $S_d(i)$ of the
$i - th$ sediment layer is:

$$\phi_{ds}(i) = \frac{\phi_0}{c}\frac{e^{-cz'(i+1)} - e^{-cz'(i)}}{S_d(i)}; \tag{9}$$





and the decompacted sediment bulk density $\rho_{ds}(i)$ of the $i-th$ sediment layer is given by:

$$\rho_{ds}(i) = \phi_{ds}(i)\rho_w + (1 - \phi_{ds}(i))\rho_S(i) \tag{10}$$

At each timestep, the removal of the top layer causes the decompaction of the underlying sediment layers. Removed sediment layers are substituted with seawater. Unloading causes an isostatic compensation that can be calculated by means of different isostatic methods. In PALEOSTRIP, two methods are implemented.

### 3.2.1 Airy local compensation

The Airy local compensation is the most used isostatic method in backstripping and backtracking. It involves a local depth compensation $Z_{airy}(i)$ for which the rocks/sediments and the underlying asthenophere are considered to be in hydrostatic equilibrium:

$$Z_{airy}(i) = S_d(i)\left(\frac{\rho_m - \rho_{ds}(i)}{\rho_m - \rho_w}\right) \tag{11}$$

This method implies that the weight of the sediments in a given grid point does not impact on the adjacent points. Therefore, in the case of a 2D transect or a 3D map, grid point result independent from each others. This is of course not realistic and the Airy local compensation should be applied preferably to the decompaction of wells only (Roberts et al., 1998).

### 3.2.2 Flexural compensation

When considering a wider area, i.e. decompacting a 2D transect or a 3D basin, a flexural compensation is required because a surface load tends to influence its surroundings. An Airy local compensation would not capture this effect and thus would overestimate the isostatic correction to be applied during the backstripping or backtracking (Roberts et al., 1998). Flexural compensation is based on the flexural strength of the lithosphere that is defined by its flexural rigidity $D$ (in N m):

$$D = \frac{ET_e^3}{12(1 - \nu^2)} \tag{12}$$

where $E$ corresponds to the Young Modulus (N/m$^2$), $T_e$ is the effective elastic thickness of the lithosphere (m) and $\nu$ is the Poisson ratio. Total 1D-flexure of the lithosphere for a line load on an infinite plate is given by the general analytical solution (Turcotte and Schubert, 2002):

$$D\frac{d^4w}{dx^4} + (\rho_m - \rho_w)gw = q \tag{13}$$

and its 2D expression:

$$D\nabla^4 w(x) = D\frac{d^4w}{dx^4} + D\frac{d^4w}{dy^4} + 2 - D\frac{d^4w}{dx^2dy^2} + (\rho_m - \rho_w)gw(x) = q \tag{14}$$

where $w$ is the deflection of the lithosphere (m), $g$ is the gravity acceleration (m/s$^2$), $q(x,y)$ is the vertical load ($N/m^2$) applied to the lithosphere and $x$ and $y$ are the coordinates in the horizontal plane. In this equation, downward deflected lithosphere is substituted with seawater $\rho_w$. The load $q(x,y)$ (in $N/m^2$) associated to each $i-th$ sediment layer is calculated as:





$$q(i) = \rho_{ds}(i)gS_d(i) \tag{15}$$

In PALEOSTRIP, a 2D and 3D finite difference versions of Eq. (13) and Eq. (14) have been implemented (e.g. Equation 9 and 10 in Wickert, 2016). They are based on Chapman (2015) `flex2d` and Cardozo (2009) `flex3dv` MATLAB codes that have been adapted to PALEOSTRIP needs. `flex3dv` routine is available at http://www.ux.uis.no/%7Enestor/Public and `flex2d` is available at https://www.jaychapman.org/matlab-programs.html. With a finite difference scheme, flexure can be computed by mean of an analytical solution (e.g. gflex, Wickert, 2016) whereas other numerical methods use the superposition of local solutions to point loads in in the wavenumber domain (e.g. Green's functions, TAFI v1.0, Jha et al., 2017), or in the spectral domain (Wienecke et al., 2007), for example. These use biharmonic equation for plate flexure with uniform elastic properties. Approaches using the convolution method also allows to use spatially variable elastic properties (e.g. Braitenberg et al., 2003, 2002). A file of spatially variable $T_e$ can be provided through the GUI (Fig. 1a, bottom). A spatially uniform $T_e$ can also be prescribed directly from the GUI. All other parameters involved in the flexural or Airy isostasy can be modified from the GUI (density constants, Poisson Ratio and Young Modulus).

### 3.3 Thermal subsidence

Thermal subsidence corresponds to a vertical contraction of the lithosphere. During a rifting phase, in a first step, the lithosphere stretches apart and thins, which causes a net increase in heat outflow towards the surface due to the upwelling of the underlying asthenosphere. The stretching is generally not uniform, but reconstructions of past stretching process require knowledge about plate tectonics strain rate (e.g. Müller et al., 2019) which is beyond the scope of our software. The first step is called "initial" subsidence. In a second step, the "thermal subsidence", the stretched lithosphere thickens due to cooling. To account for those effects during backstripping, PALEOSTRIP adopts the 1D-thermal subsidence model from McKenzie (1978) that assumes an instantaneous stretching (syn-rift) and a single rifting phase. The initial subsidence $Z_{init}$ is given by:

$$Z_{init} = \frac{T_{IL}\left[(\rho_m - \rho_c)\frac{T_{IC}}{T_{IL}}\left(1 - \frac{\alpha_v T_m T_{IC}}{2T_{IL}}\right) - \frac{\rho_m \alpha_v T_m}{2}\right]\left(1 - \frac{1}{\beta}\right)}{\rho_m(1 - \alpha_v T_a) - \rho_w} \tag{16}$$

where $T_{IL}$ and $T_{IC}$ are the initial lithospheric and crustal thicknesses at the beginning of rifting, $\beta$ is the stretching factor, $\alpha_v$ is the coefficient of thermal expansion, $\rho_c$ is the crustal density, $\rho_m$ is the mantle density. In this equation, the subsidence is isostatically compensated by water ($\rho_w$) using the Airy local compensation. This 1D instantaneous model can be improved by adopting a time-evolving approach to the extension, such as for example Jarvis and McKenzie (1980). However, comparison between Jarvis and McKenzie (1980) and McKenzie (1978) revealed that the two models show no or only little discrepancy if the duration of extension, given the time required to extend by a factor $\beta$, is about $60/\beta^2$ Ma if $\beta \leq 2$ or $60(1 - 1/\beta)^2$ Ma if $\beta \geq 2$ (Jarvis and McKenzie, 1980). During the second step, the thermal subsidence occurs after the end of rifting (post-rift) and accounts for the vertical thermal conduction which takes the shape of an exponential function decaying with time:

$$Z_{thermal}(t) = E_0 \frac{\beta}{\pi} sin\left(\frac{\pi}{\beta}\right)\left(1 - exp\left(-\frac{t}{\tau}\right)\right) \tag{17}$$





where $t$ corresponds to the time elapsed since the end of rifting expressed in seconds (example: time of backtracked horizon = 14 Ma; time since the end of rifting = 76 Ma; $t = (76 - 14) * 3600 * 24 * 365 * 10^6$ ).

$$E_0 = \frac{4\rho_m \alpha_v T_m T_{IL}}{\pi^2 (\rho_m - \rho_w)}, \ \ and \ \ \tau = \frac{T_{IL}^2}{\pi^2 \kappa} \tag{18}$$

where $\kappa$ is the thermal diffusivity. The total thermal subsidence is given by:

$$Z_{tot\_thermal} = Z_{init} + Z_{thermal} \tag{19}$$

In PALEOSTRIP, during the backtracking procedure, paleo-water depths are corrected by removing the increment of thermal subsidence between each timestep and present-day. This is because backtracking is an iterative process departing from a known state, i.e. present-day. Because $Z_{init}$ is constant in time, it cancels out, and the thermal subsidence correction is given by:

$$\Delta Z_{tot\_thermal}(t) = (Z_{init} + Z_{thermal}(t)) - (Z_{init} + Z_{thermal}(0)) \tag{20}$$

and thus,

$$\Delta Z_{tot\_thermal}(t) = Z_{thermal}(t) - Z_{thermal}(0) \tag{21}$$

where $t$ varies from present to past in this equation following backstripping procedure, i.e., $t$ is an age older than present-day (0). $\Delta Z_{tot\_thermal}$ is thus positive with $Z_{thermal}$ being larger for younger times, i.e. larger time ellpsed since the end of rifting.

Note that by applying the 1D model from McKenzie (1978) to 2D transects and 3D maps, it is assumed that no horizontal heat advection occurs. From the GUI tab interface (Fig. 1b), almost all parameters involved in the thermal subsidence can be

modified (Age of end of rifting, $T_{IL}$ and $T_{IC}$, $\alpha_v$, $\kappa$). The stretching factor $\beta$, can be formulated through several methods:

(1) prescribe a constant and uniform $\beta$ factor that will be used at each timestep of backtracking to calculate the thermal subsidence.

(2) linearly interpolate (in the x direction for 3D maps) between two prescribed constant and uniform stretching factors, $\beta_1$ and $\beta_2$

(3) input a user-based constant but spatially variable $\beta$ factor, in one direction (x grid), to compute spatially variable thermal subsidence for a 2D transect or input a 2D (x,y grid) file to compute spatially variable thermal subsidence for 3D maps.

### 3.4   Sea Level correction

Sea level has been varying with time due to plate tectonics changing the dimensions of ocean basins (e.g. Müller et al., 2018a) and due to continental ice storage within ice sheets, ice caps and glaciers during cold periods, as for example during the second

half of the Cenozoic (last 34 Ma, e.g. Miller et al., 2020). Classically, only eustatic sea level changes have been considered for correcting the subsidence or the paleo-water depth. For the last 34 Ma, eustatic sea level is usually defined relatively to the present-day total ocean area ($362.15 \ 10^6$ km$^2$) because it is assumed that this area evolved only a little and not enough





to significantly alter this number. However, for time periods older than that, eustatic sea level changes have to be expressed according to the ocean area of the time. Considering only eustatic sea level changes is highly approximated since in the more

recent history, sea level has also been influenced by ice sheets growth and decay, leading to sea level variations of growing amplitude over the past 34 Ma (e.g. Miller et al., 2020). The variations induced by glaciations have much shorter timescales, i.e. $10 - 10^5$ yrs, compared with those induced by plate tectonics ($10^5 - 10^6$ yrs). Associated sea level changes are not spatially uniform and induce changes in the Earth's gravity field (e.g. Tamisiea et al., 2001) and regional self-gravitating sea level changes can substantially vary from eustasy (e.g. Tamisiea et al., 2001; Clark et al., 2002; Milne and Mitrovica, 2008).

In PALEOSTRIP sea level ($SL$, in m) is corrected as for the thermal subsidence:

$$\Delta SL(t) = SL(t) - SL(0) \tag{22}$$

where $t$ varies from present to past in this equation following backtracking procedure, i.e., $t$ is an age older than present-day(0). $\Delta SL$ is expressed relative to present and it can therefore assume positive or negative values, i.e. induce an uplift or a subsidence. Most of the sea level variations timeseries found in literature already correspond to variations relative to present,i.e., already

are in the form of Eq. 22. Thus in order to avoid confusion for the user, three different ways of correcting water depth with sea level changes have been implemented within PALEOSTRIP and can be managed through the GUI (Fig. 1c):

(1) prescribing constant and uniform sea level correction $\Delta SL$ applied at each timestep of the backtracking (ex: correction at a time where sea level is lower by 100 meters relative to present is prescribed as $-100$ in the GUI);

(2) applying a spatially uniform but time-varying sea level correction based on timeseries from Haq et al. (1987) or from
Miller et al. (2020) (implemented within PALEOSTRIP);

(3) applying a user-provided timeseries or a constant in time but spatially varying map (x,y, z) of sea level changes relative to present.

The last option allows to prescribe sea level changes calculated with Glacio-Isostatic Adjustment models (e.g. SELEN[4] Spada and Melini, 2019) to account for regional self-gravitating effects of ice sheets growth and decay. $\Delta SL$ is used to compute

the water-load correction to adjust the water depth $WD$. In Eq. 3 water load due to sea level change is described based on Airy local compensation. In PALEOSTRIP, the water-load due to sea level change is computed using the isostatic method previously selected to carry out decompaction (see Sect. 3.2). Note that if isostasy is deactivated, no water-load correction is computed.

## 3.5 Dynamic topography

Mantle dynamics generate flows that cause time-varying surface topography and bathymetry deformations, which are called

"dynamic topography". The timescale at which the mantle flow produces dynamic topography, occurring at long wavelength ($\approx 5\,000$ to $10\,000$ km Hoggard et al., 2016). The current mantle- driven dynamic topography was revealed by estimating the residual topography, obtained after removing the isostatic topography, generated by thickness and density contrasts within the lithosphere the isostatic effect of the lithosphere calculated using seismic data compilations (e.g. Flament et al., 2013; Hoggard





et al., 2016). Continental margins stratigraphic observations revealed that the dynamic topography evolves quickly with time,
thus potentially impacting on long-term climate evolution (Hoggard et al., 2016). Austermann et al. (2017) showed that the
magnitude of past interglacial sea level proxies partly results from dynamic topography and suggested correcting sea level
proxies before inferring the relative contribution of past ice sheet to the sea level changes at that time. Furthermore, simulated
Pliocene dynamic topography changes accounted for in Antarctic ice sheet simulations revealed that ice sheet stability is
highly influenced by mantle dynamics that create or cancel pinning areas at the surface (Austermann et al., 2015). Thus
past reconstructions of continental morphology and shallow margins should account for past evolution of dynamic topography.
Müller et al. (2018a) recently provided timeslices of reconstructed dynamic topography over the past 240 Ma, which constitutes
a strong basis to calculate a dynamic topography correction.

PALEOSTRIP provides three ways to correct for dynamic topography changes through its GUI (Fig. 1d):

(1) prescribing a uniform and constant dynamic topography change relative to present-day that will be applied to each
timestep of the backtracking procedure;

(2) using a user-based spatially uniform timeseries of dynamic topography changes relative to present;

(3) using user-based 2D maps of dynamic topography changes relative to present, e.g. Müller et al. (2018a); Note that Müller
    et al. (2018a) is not implemented within PALESOTRIP and requires some post-processing to be adjusted to the area of
    interest before being passed through the GUI.

The correction is calculated as for sea level changes (Eq. 22):

$$\Delta DynT(t) = DynT(t) - DynT(0) \tag{23}$$

where $t$ varies from present to past in this equation following backstripping procedure, i.e., $t$ is an age older than present (0).
$\Delta DynT$ is expressed relative to present, and since dynamic topography is not spatially uniform at global level, it can therefore
assume positive or negative values, i.e. induce an uplift or a subsidence.

**4   PALEOSTRIP Grid interpolation**

PALEOSTRIP handles 1D data (drill sites), 2D data (transects) and 3D data (grids). All calculations, except flexure, are
performed on the grid or array of original input data. For 2D and 3D data, all input horizon depths have, however, to be on
the same grid/array, and have a regular horizontal grid resolution. For example, 2D transects must have horizons depths of the
same length and with a regular horizontal resolution. 3D maps can be provided as an irregular polygon and the horizontal grid
resolution $dx$ and $dy$ can differ but have to be constant and all horizon depths must be provided on the same exact polygon (see
examples in Sect. 6). At last, grid horizontal resolution $dx$ and $dy$ must be integers (e.g. dx= 10 km and not dx= 4.3 km).




## 4.1 2D data: transects

Input data for transects must be provided on a regular array for each horizon. For the need of flexure calculation, the original sediment loads array is extended (duplication of last values at both edges) to about 80 % of its length from both edges (Fig. 4a). The sediment loads are then placed at the center of an expanded domain (three times larger than original array length) to avoid edge effects on flexure correction. After flexure calculation, the correction is extracted and relocated on the original array domain. All the other backtracking calculations occur on the original input array. Note that spatially variable lithospheric elastic thickness is also interpolated and extrapolated following this procedure.

## 4.2 3D data: maps

Input data need to be provided on the same polygon (irregular domain edges) or grid (rectangular domain) with a constant $dx$ and $dy$ (which can be different). Input data are read as scattered data, and not as gridded data. This facilitates all computations and avoids unnecessary duplication of routines for 1D, 2D or 3D cases. For the need of flexure, 3D data are interpolated (preserving their original horizontal resolution) on a regular rectangular grid. The original input grid is expanded (extrapolation of the last values from the edges) to about 30% from all sides of the grid. The sediment loads are then placed at the center of an expanded domain (twice larger than original grid dimension) to avoid edge effects on flexure correction (Fig. 4b). After flexure calculation, the correction is extracted and relocated on the original grid domain. All the other backtracking calculations occur on the original input grid. Note that spatially variable lithospheric elastic thickness is also interpolated and extrapolated following this procedure.

## 5 PALEOSTRIP validation

We backtrack a 2D transect with PALEOSTRIP and with Flex-Decomp (Kusznir et al., 1995) to validate the results. The transect used in the case study is a revised version of the one studied by De Santis et al. (1999), the BGR80-007 seismic profile. The transect BGR80-007 is composed of 9 identified seismic stratigraphic unconformities. The transect is about 250 km long, broadly oriented North-South and is located in the Eastern Ross Sea (Fig. 5). The set of initial data is composed of ten files: the actual bathymetry of the Ross Sea, present-day depth of the nine seismic unconformities including present-day depth of the basement below (Fig. 6a).

The eleventh file contains the lithological parameters of the layers to be decompacted, as well as other parameters needed by PALEOSTRIP, excluding present-day bathymetry: LAYER is the layer number (1 to N, from bottom to top); POROSITY is the deposition porosity (unitless), DEC CON (1/KM) correponds to the porosity decompaction coefficient, MAT DEN (KG/M3) corresponds to the compacted sediment density; AGE BASE (Million years ago, Ma) is the age of horizons and NAME is the string name of horizons. The parameters are taken from the study of De Santis et al. (1999) and the lithological parameter file follows this format:

```
NUMBER OF LAYERS =            9
```





| LAYER | POROSITY | DEC CON (1/KM) | MAT DEN (KG / M3) | AGE BASE (MA) | NAME |
|-------|----------|---------|---------|---------|------|
| 1 | 0.4900 | 0.2700 | 2680.00 | 95.00 | basement |
| 2 | 0.4500 | 0.4500 | 2680.00 | 26.00 | rsu_6 |
| 3 | 0.4500 | 0.4500 | 2680.00 | 24.90 | rsu_5b |
| 4 | 0.4500 | 0.4500 | 2680.00 | 19.70 | rsu_5a |
| 5 | 0.4500 | 0.4500 | 2680.00 | 18.00 | rsu_5 |
| 6 | 0.4500 | 0.4500 | 2680.00 | 14.20 | rsu_4 |
| 7 | 0.4500 | 0.4500 | 2680.00 | 10.00 | rsu_3 |
| 8 | 0.4500 | 0.4500 | 2680.00 | 4.00 | rsu_2 |
| 9 | 0.4500 | 0.4500 | 2680.00 | 0.60 | rsu_1 |

To facilitates the comparison, thermal subsidence, sea level and dynamic topography are switched off both in PALEOSTRIP and in Flex-Dedcomp. We compare the final backtracked depths of the basement using Airy local isostasy and flexural isostasy with different lithospheric elastic thicknesses (Fig. 6b). Match between PALEOSTRIP and Flex-Decomp is very good and persistent discrepancies (a few tens to a hundred of meters) are likely due to (1) the different ways of computing flexural isostasy, in the spectral domain for Flex-Decomp and with finite difference for PALEOSTRIP and (2) to re-interpolation of the load to a different resolution in Flex-Decomp (no reinterpolation in PALEOSTRIP) and (3) to different extrapolation of the load outside of the original transect length to avoid edge effect on flexure. In PALEOSTRIP, the last point of the transects at both edges is duplicated to extend the original length of about 80% at both sides (Fig. 4a). We also compare PALEOSTRIP backtracked results with those using the analytic solution from TAFI v1.0 (Jha et al., 2017) and implemented within PALEOSTRIP for the need of comparison (Fig. 6c). Similarly to the comparison with Flex-Decomp, we only account for isostasy and the other processes are switched off. Comparison between PALEOSTRIP (finite difference scheme) and TAFI v1.0 reveal almost identical results.

Finally, we test the isostasy and thermal subsidence model by comparing those obtained by PALEOSTRIP and Flex-Decomp. De Santis et al. (1999) originally used Flex-Decomp with $\beta$=2, with the age of rifting set to 85 Ma and various lithospheric elastic thickness values to restore paleobathymetries. We use the same parameters on this revised transect both in Flex-Decomp and in PALEOSTRIP. Match between PALEOSTRIP and Flex-Decomp is very good (Fig. 6d).

## 6 Case study: example of the Ross Sea

PALEOSTRIP GUI presents a Plot & Save interface to support each steps of backtracking: the user can plot initial input data, backtracked data, calculated intermediate variables relevant to the backtracking process and save them to ASCII files (Fig. 7). The user can also extract some 2D transects or 1D well from 3D backtracked maps or 2D transects, plot and save them



to ASCII files. In the following, we provide some case studies to illustrate the possibilities of PALEOSTRIP. Note that the physical parameters, sea level correction or thermal subsidence related variables are not tuned since the aim of the examples is to illustrate the physics of PALEOSTRIP rather than to provide a realistic reconstruction of the paleo-bathymetries of this area.

All the cases are taken from the continental margin in the Western Pacific Sector of Antarctica, in the Ross Sea (Fig. 5).
390   Sediment layers mostly accumulated after the main rifting phases that occurred in this area between 95 and 79 Ma (Stock and Cande, 2002; Decesari et al., 2007; Siddoway et al., 2004). **Ross Sea stratigraphic data are currently being revised and differ from ANTOSTRAT data** on which the following 3D example is based. New reconstructed Ross Sea bathymetry using revised data will be the object of a specific contribution. ANTOSTRAT data have been recently used in pan-Antarctic reconstructions of past topographies and bathymetries (Paxman et al., 2019; Hochmuth et al., 2020).

395

## 6.1   Well (1D): Deep Sea Drilling Project DSDP site 270 - Ross Sea

In the last example, we decompact the drilling site DSDP-270 Hayes and Frakes (1975) from the Western Ross Sea (Fig. 5). It has two main identified seismic unconformities above the basement. Lithological parameters are taken from De Santis et al. (1999) and the lithological parameters file (see Sect. 5) is:

```
400   NUMBER OF LAYERS =              3

      LAYER   POROSITY    DEC CON    MAT DEN   AGE BASE      NAME
                          (1/KM)    (KG / MC)    (MA)

405   1     0.4500     0.2700     2680.00     95.00     basamento
      2     0.4500     0.4500     2680.00     21.00     rsu_5
      3     0.4500     0.4500     2680.00     19.00     rsu_4a
```

Backtracking is performed using Airy local isostasy since flexure cannot be applied to 1D drilling sites. We also add thermal subsidence in order to illlustrate the difference in backtracked depths of those unconformities (Fig. 8). The layout of Fig. 8 is
410   not the layout of PALEOSTRIP but results have been assembled to highlight the impact of thermal subsidence on backtracked depths. As for 2D transects, all intermediate variables and input conditions can be plotted and saved.

## 6.2   Map (3D): ANTOSTRAT data

In this example we backtrack the Mid-Miocene paleo-bathymetry ( 14 Ma) across the Ross Sea. The initial grid is an irregular polygon. The set of initial data is composed of three files: the actual bathymetry of the Ross Sea, present-day depth of Mid-
415   Miocene unconformity and presumed present-day depth of the basement below (Fig. 9).

The fifth file contains the lithological parameters of the layers to be decompacted (see Sect. 5) and are taken from De Santis et al. (1999) and the lithological parameters file is:



```
NUMBER OF LAYERS =          2

LAYER   POROSITY    DEC CON    MAT DEN   AGE BASE   NAME
        ( % )       (1/KM)    (KG / CC)   (MA)

 1     0.450      0.450      2680.0     95.000    Basement
 2     0.450      0.450      2680.0     14.200      RSU4
```

PALEOSTRIP is run several times to add one component at once amongst, isostasy, thermal subsidence, sea level correction and dynamic topography correction. Thanks to this approach, the user can perform ensembles of simulations to retrieve sound statistics about the model parameter space. Two series are shown, one using Airy isostatic correction (Fig. 10), and the other using flexural isostatic correction (Fig. 11). Paleo-bathymetry retrieved with flexural isostasy produces a quite different morphology from the one computed using Airy isostasy, as already observed by Roberts et al. (1998). Sea level and dynamic topography (Fig. A1b) corrections are not big enough to produce a significant change of the overall morphology. However, they matter for the bathymetric highs, and especially, the shallowest one, as few tens of meters can uplift those highs above sea level. PALEOSTRIP allows the user to plot different variables amongst initial input data and computed quantities, such as isopach, density, porosity or isostatic correction, so the user can check and separate the various processes to perform a detailed analysis of their impact on the paleo-bathymetric reconstruction (Fig. A1).

## 7 Conclusions

PALEOSTRIP is one of the first open-source 3D backtracking software. It can process 1D drilling sites, 2D transects and 3D maps. It allows to separate the various processes involved in the backtracking procedure. Thanks to this approach, the user can perform ensembles of simulations to retrieve sound statistics about the model parameter space. PALEOSTRIP has been designed to be modular, to allow user to insert their own modifications. The code is documented and implementation of new modules would only require a minor work.

*Code and data availability.* The version of the code and example data used in this manuscript are available on Zenodo https://doi.org/10.5281/zenodo.4607300 or on github https://github.com/flocolleoni/PALEOSTRIPv1.0.

## Appendix A: Case studies: settings

## A1 Table of Physical parameters

In all the examples provided, we used PALEOSTRIP default values automatically inserted within GUI (Table 1).



## A2 PALEOSTRIP workflow

## A3 Backtracked intermediate variables

Here we plot some of the intermediate physical variables calculated during the backtracking procedure and available for plotting through PALEOSTRIP GUI: sediment isopachs, decompacted density, decompacted porosity, isostatic correction and dynamic
topography (Fig. A1). Thermal subsidence is also available for plotting but since here we employ a spatially uniform $\beta$ value, thermal subsidence is consequently spatially uniform over the domain and thus we do not display it.

*Author contributions.* F.C. developed PALEOSTRIP code. L.D.S. and R.G. provided input data for the case studies. All the authors contributed to discussions about numerical developments of PALEOSTRIP and writing of the manuscript.

*Competing interests.* The authors declare no competing interests.

*Acknowledgements.* This work is supported by the PNRA national Italian projects: PNRA18_00002, "Onset of Antarctic Ice Sheet Vulnerability to Oceanic conditions (ANTIPODE)" PNRA16_00016, "West Antarctic Ice Sheet History from Slope Processes–Eastern Ross Sea (WHISPERS)" and by the MAE bilateral Italy-US project US16GR04, "Global Sea Level rise & Antarctic Ice Sheet Stability predictions: guessing future by learning from past(GLSAISS)".





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



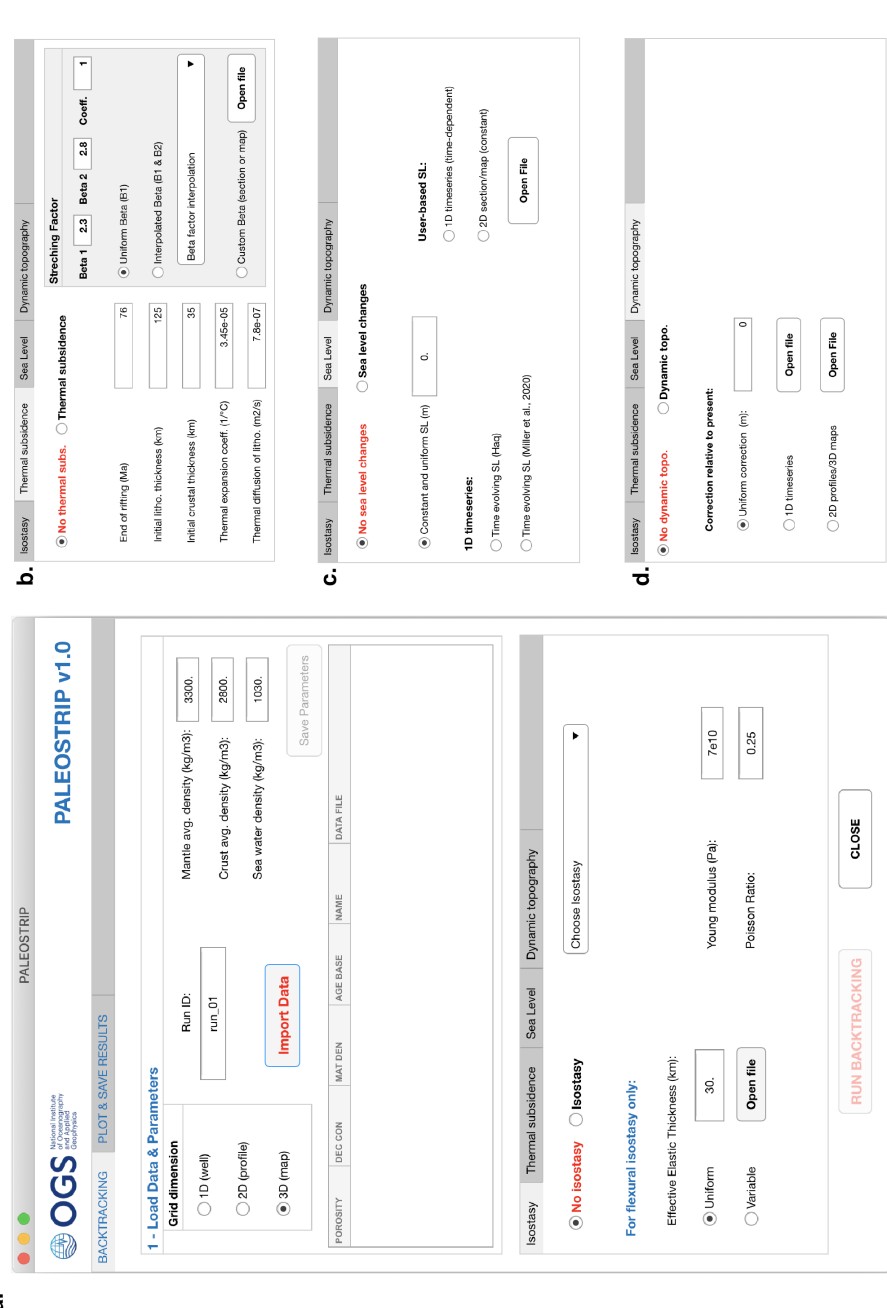

**Figure 1.** PALEOSTRIP Application GUI: a. Main PALEOSTRIP GUI; b. tab interface to set thermal subsidence (bottom of main GUI); c. tab interface to set Sea Level corrections; d. tab interface to set Dynamic topography correction.



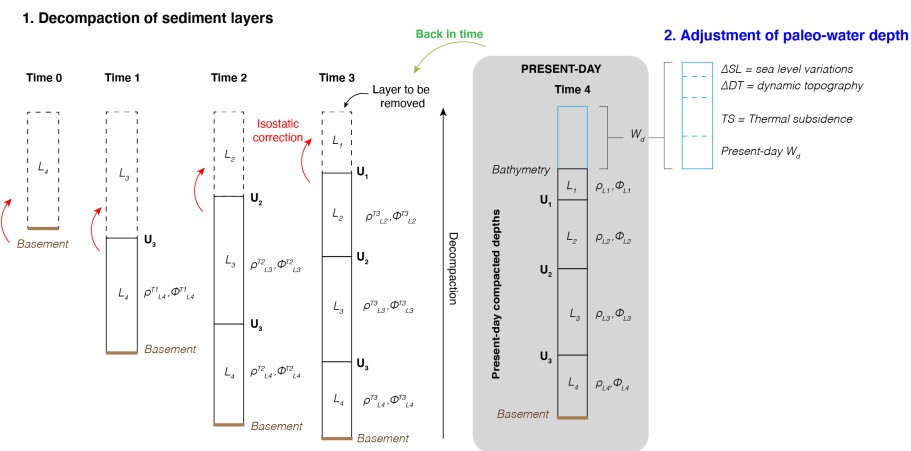

**Figure 2.** Backstripping/Backtracking procedure: sediment layers $L$ are separated by well-defined seismic or lithological unconformities $U$. All the sediment layers have different lithological properties defined by their porosity $\phi$ and density $\rho$. Those two properties changes during the decompaction process, departing from present-day depth of sediment layers, as buried sediments are unloaded from the upper sediment layers (isostatic correction). At this point, the procedure differs: if the aim is backstripping, then the depth of the sediment layers at each time step is adjusted with prescribed time-history of water depth changes (Eq. 1), if the aim is backtracking, the water depth at each time step is calculated using time-varying thermal subsidence model (Eq. 3). PALEOSTRIP computes backtracking.





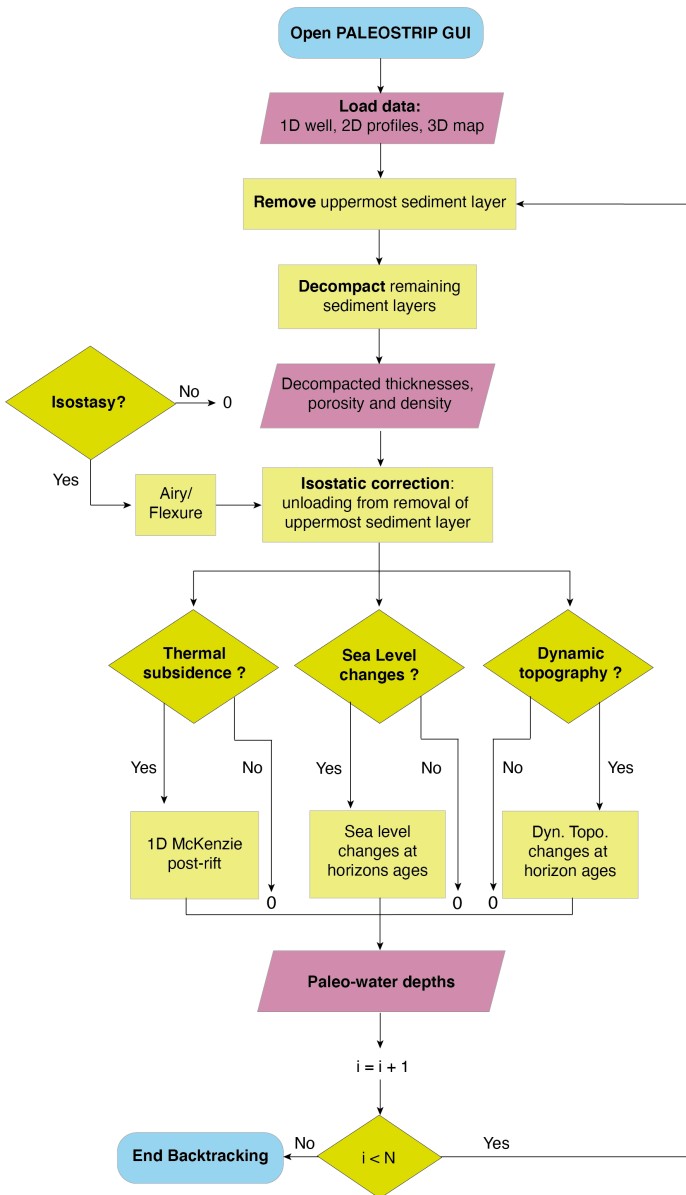

**Figure 3.** Workflow of the code implemented in PALEOSTRIP illustrating the various steps of computation according to selected options in the GUI. N sediment layers are decompacted during the backtracking procedure. Smooth blue rectangles correspond to start and end of the backtracking run. Light red parallelograms indicate input and output data. Light green rectangles are intermediate steps in the backtracking computation. Green diamonds correspond to options selected by the user in the GUI. If isostasy, or thermal subsidence, or sea level changes, or dynamic topography are switched off, the corresponding correction equals zero.



**a. 2D array expansion for flexural isostasy**

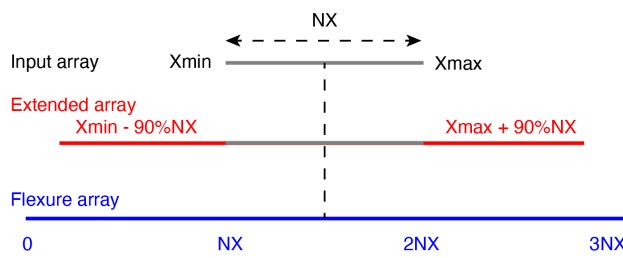

**b. 3D grid interpolation for flexural isostasy**

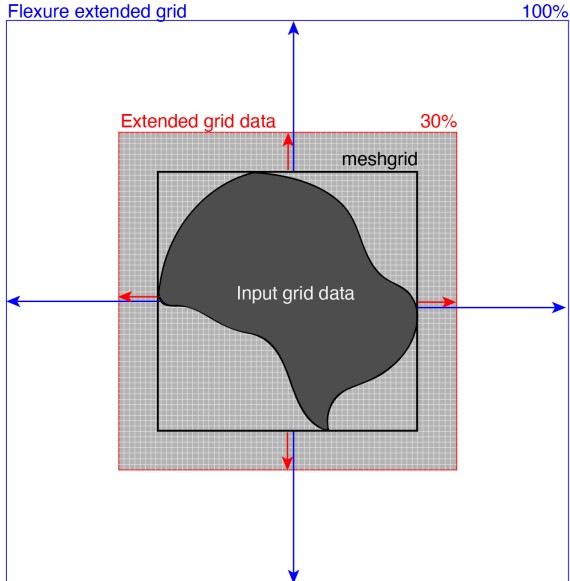

**Figure 4.** 2D and 3D domain expansion for computation of flexural isostasy: a. In case of 2D vertical sections, input data are extended with the edges values for 90% of the initial total array length (NX). This extended array is placed at the centered of a null array three times larger than the initial input data array.; b. In case of 3D maps, input data are extrapolated with a nearest neighbor algorithm on a mesh grid (black) to obtain a structured squared grid if the initial data are given on an irregular polygon. This mesh grid is extended by 30% on all edges (red) and is placed at the centered of a grid twice as large as the mesh grid obtained from initial input data (blue).



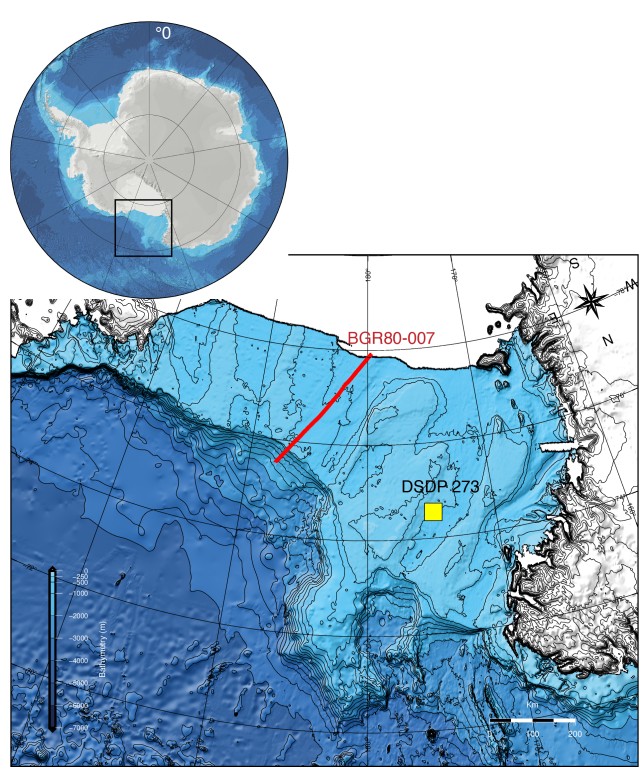

**Figure 5.** Close-up of the Ross Sea bathymetry from IBCSO (Arndt et al., 2013) and its location in Antarctica. Most of the ice-free bathymetry is backtracked as a case studies below. The BGR80-007 2D marine seismic transect used for validation is indicated with a red line and the DSDP 273 site location, backtracked in the case studies below, is indicated with a yellow square.



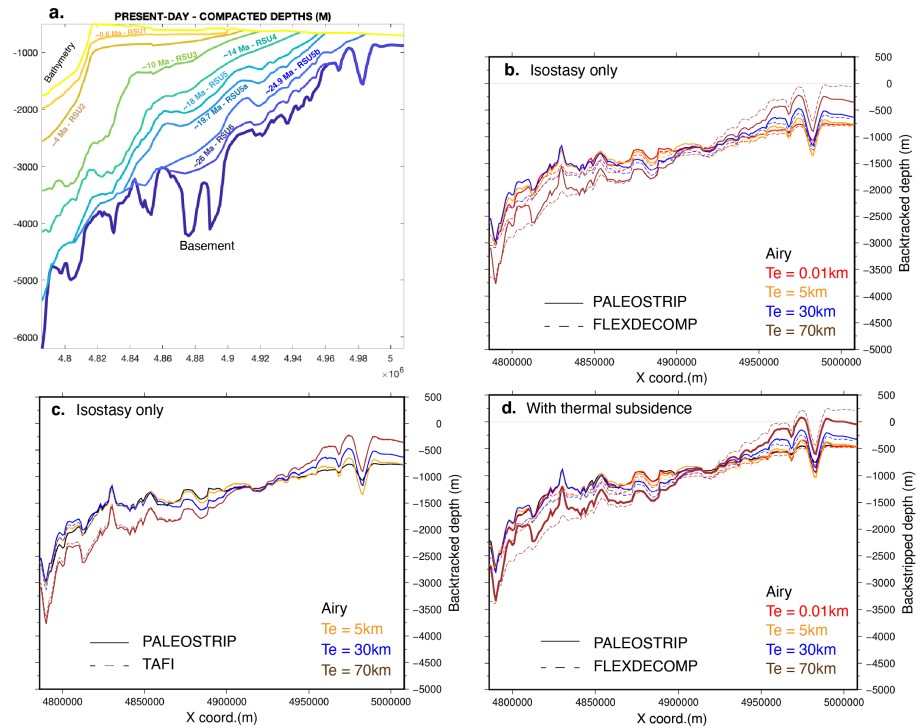

**Figure 6.** 2D transect BGR80-007 case study: a. Input present-day depths (m) of bathymetry (yellow), and the nine seismic unconformities, amongst which, the mid-Miocene Ross Sea Unconformity 4 (RSU4) and basement (thick blue line). Layout is from PALEOSTRIP Plot GUI. Comparison of backtracked basement depths for different but spatially uniform lithospheric elastic thicknesses (Te): b. between PA-LEOSTRIP finite difference isostasy and Flex-Decomp (Kusznir et al., 1995) Fast Fourier Transform (wavenumber-based) isostasy model; c. between PALEOSTRIP finite difference isostasy and TAFI (Jha et al., 2017) Green's function spectral isostasy model; d. Same as b. but accounting for thermal subsidence correction using an age of Rift = 85 Ma and a uniform $\beta = 2$.



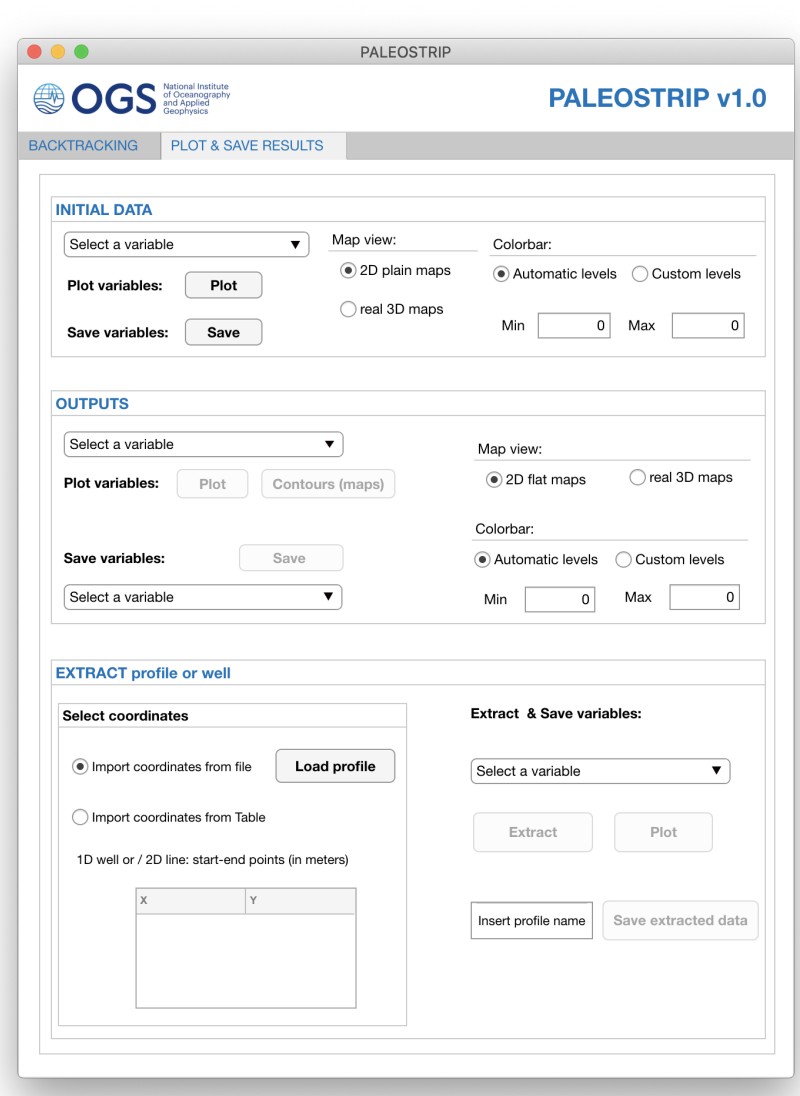

**Figure 7.** PALEOSTRIP Application GUI: Plot & Save Results interface.



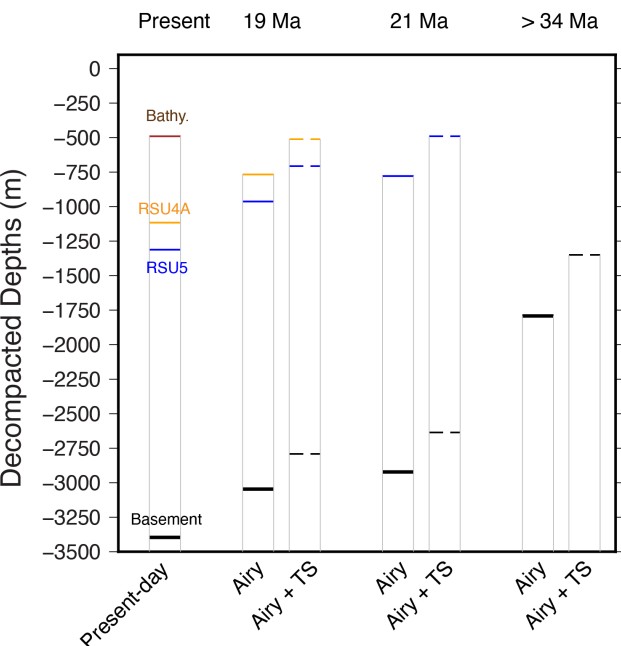

**Figure 8.** Backtracking steps for Western Ross Sea DSDP 270 well site. For present-day: present-day depths of basement (black), Ross Sea unconformity 5 ( blue), Ross Sea Unconformity 4A (orange) and bathymetry (brown); at time of RSU4A (19 Ma), at time of RSU5 (21 Ma) and at time before glacial sediment deposition (> 34 Ma). For each time step, two backtracking are shown, one accounting for Airy isostatic correction only, and one accounting for Airy isostatic correction and thermal subsidence (dashed). Table 1 in Appendix A.



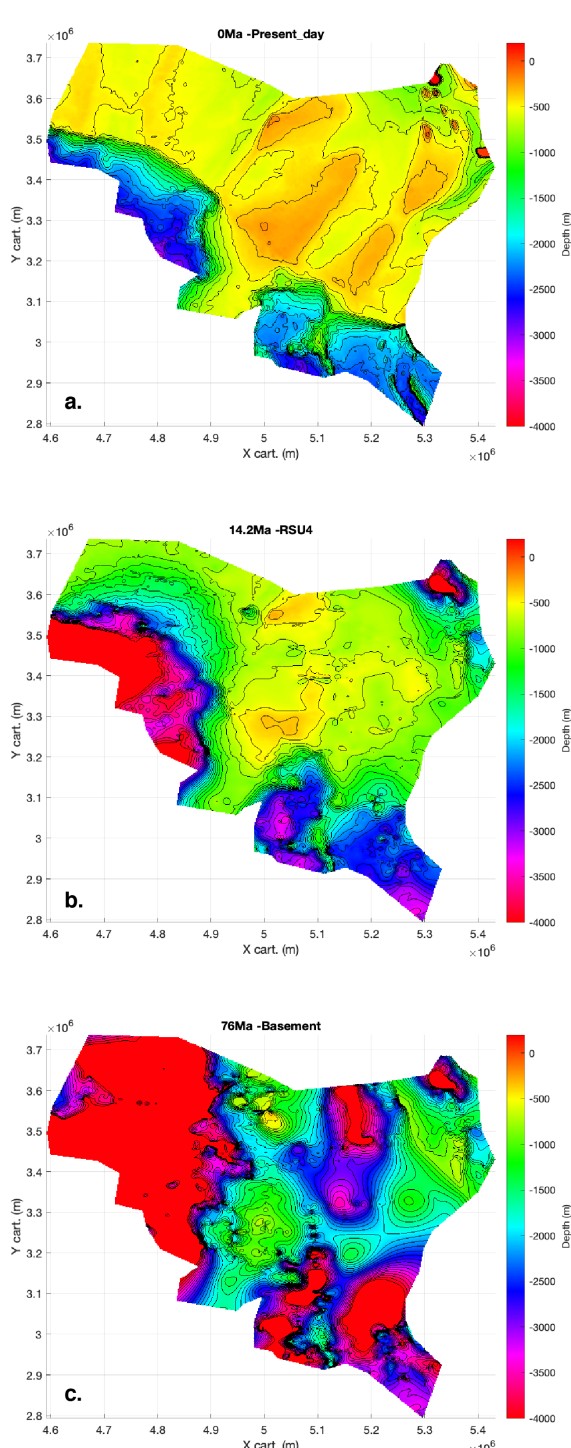

**Figure 9.** Initial input present-day depths (m): a. bathymetry, b. mid-Miocene unconformity RSU4 and c. basement. Layout is from PALE-OSTRIP plotting GUI. Colorscale is saturated below -4000 meters and above 200 meters.

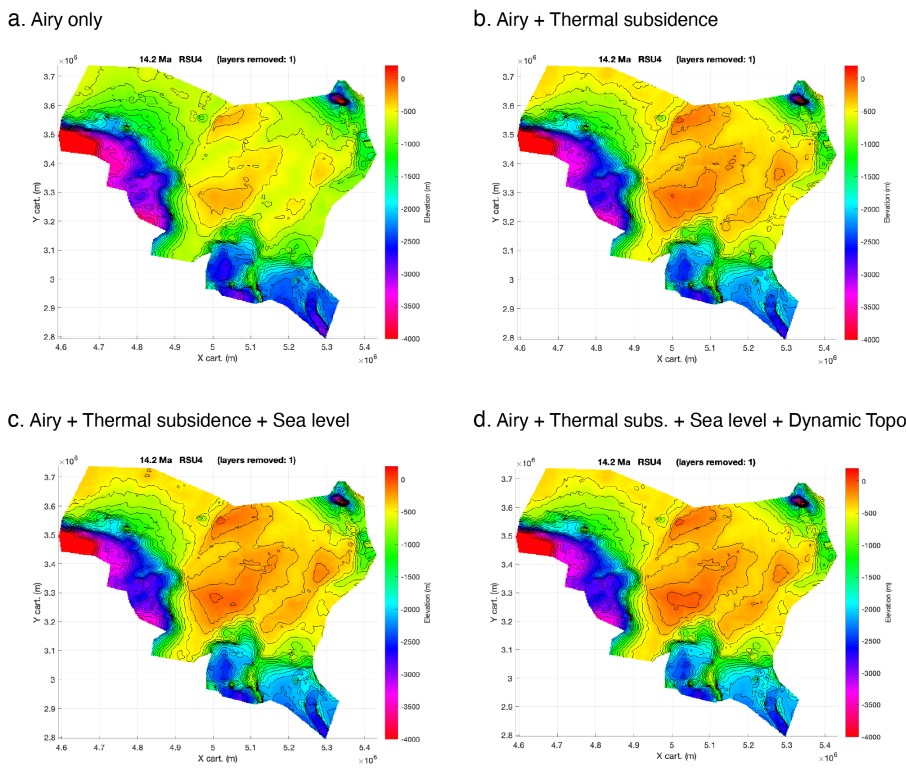

**Figure 10.** Backtracked Mid-Miocene bathymetries at about 14 Ma: a. corrected only for Airy isostasy; b. same as a. but including thermal subsidence correction with a spatially uniform $\beta = 2.1$ and the end of rifting sets at 76 Ma; c. same as b. but including eustatic sea level correction based on Miller et al. (2020) $\delta^{18}O$-derived reconstruction; d. same as c. but including dynamic topography correction based on Müller et al. (2018a) geodynamical model M1 (following Hochmuth et al., 2020). Colorscale is the same as in Fig. 9. For all parameters used in those example see Table 1.

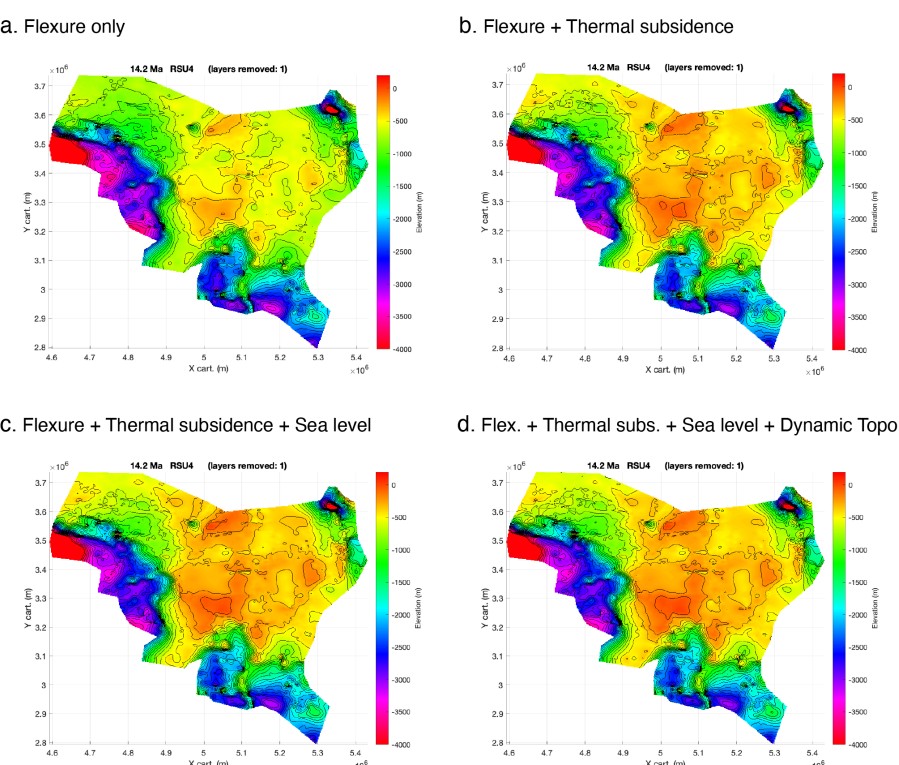

**Figure 11.** Backtracked Mid-Miocene bathymetries at about 14 Ma. Same as Fig. 10 but using flexural isostasy with spatially variable lithospheric effective elastic thickness from Chen et al. (2018). Colorscale is the same as in Fig. 9. For all parameters used in those example see Table 1 in Appendix A.



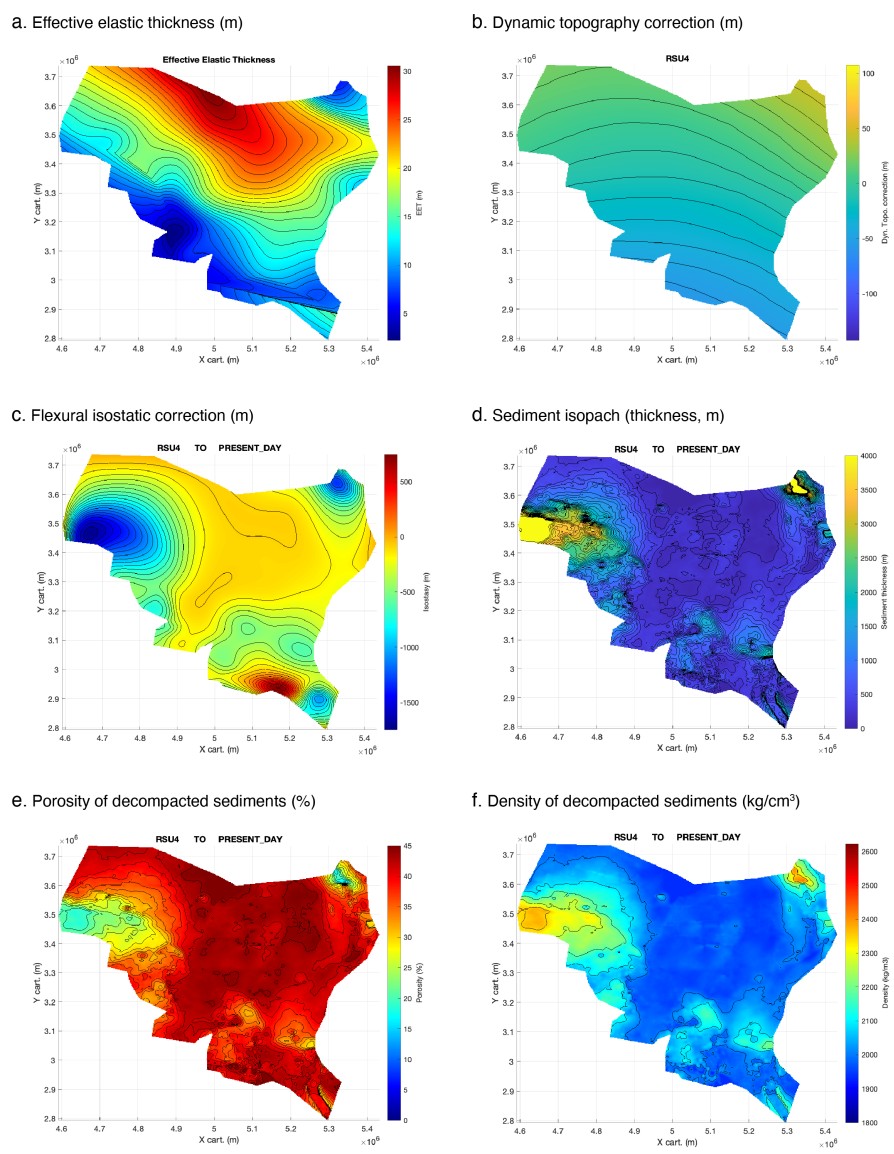

**Figure A1.** Example of input data and related backtracked sediment variables at Mid-Miocene: a. spatially variable lithospheric elastic thickness (Chen et al., 2018); b. Dynamic topography correction (Müller et al., 2018b). Both a. and b. are user-provided input to PALEOSTRIP; c. Flexural isostatic correction calculated using a.; d. Decompacted isopach (m); e. Decompacted porosity (%); f. Decompacted density $(kg/cm^3)$





**Table 1.** Physical parameters values used in the case studies.

| **Isostasy:** | |
| --- | --- |
| Young modulus (Pa) | 7e10 |
| Poisson Ratio | 0.25 |
| **Thermal Subsidence:** | |
| End of rifting (Ma) | 76 |
| Initial lithospheric thickness (km) | 125 |
| Initial crustal thickness (km) | 35 |
| Thermal expansion coefficient (1/deg C) | 3.45e-05 |
| Thermal diffusion of lithosphere (m$^2$/s) | 3.45e-05 |
| Streching Factor | 2.1 |
| **Sea Level correction:** | |
| 1D timeseries | Miller et al. (2020) |
| **Dynamic Topography:** | |
| 3D maps | time-evolving and spatially varying Müller et al. (2018b) |