# Peer review of "PALEOSTRIPv1.0 - a user-friendly 3D backtracking software to reconstruct paleo-bathymetries"

_Geoscientific Model Development, 2021_

## Referee Comment (RC1)

**Review of paper by Colleoni et al.: *PALEOSTRIPv1.0 – a user-friendly 3D backtracking software to reconstruct paleo-bathymetries;* submitted to GMD (MS No. gmd-2021-78)**

**General comments:**

This paper presents the rationale, methodology, physical/geophysical background and parameterization of new open-source software to calculate paleo-water depths in a 1D, 2D and 3D backtracking/backstripping procedure. The development of a relatively user-friendly open-source software for paleobathymetry is timely as the relevance for improved paleo-water depths for different time slices of the past geological history of ocean basins and continental margins is largely increased for urgently needed improved paleo-ice sheet and paleo-climate modelling results. Previously available software solutions have been lacking various aspects, such as full 3D flexural response, implementation of mantle-driven dynamic topography, sensitivity testing, open-access/open-source, and user-friendliness. All aspects are absolutely necessary to a research community-wide acceptance and usage of this software. The paper is well structured and written, and it contains the necessary and informative tables and figures. The methods and mathematical background are absolutely sound, and previously developed procedures are correctly cited. I recommend the paper to be accepted for publication after the following minor issues are addressed.

**Specific comments and suggestions** (numbers refer to line numbers of manuscript):

1) In general: Difference between backtracking and backstripping?  I think that this is rather a semantic difference. In both approaches, the same fundamental principles and equation systems are used. It's only a matter which of the variables should be solved or kept constant first: Either assuming a fixed paleo-water depth and then solve for decompaction and subsidence, or implying an assumed subsidence and decompaction for obtaining a paleo-water depth. I suggest to generally use the term backstripping for either approach as this term is widely known and recognised in the geoscience community.

3) Introduction: I suggest to add the following two references as previous examples of 3D backstripping cases from the SW African and Brazilian margins to the Introduction: Dressel et al. (Marine & Petroleum Geology, 2015) and Dressel et al. (Tectonophysics, 2017).

2) Chapter 3: The authors describe with great clarity all necessary steps for backtracking/backstripping (decompaction, isostatic correction, thermal subsidence, sea-level, dynamic topography), but one important aspect is missing which is in particular important for polar continental shelves. That is the reconstruction of sediments that were eroded/truncated by grounded ice-sheet advances. Of course, reconstructing such eroded sediments above truncational unconformities or the seafloor (in case where dipping sequences pinch out) requires assumptions on pre-erosional sedimentation rates, but it would be useful if the authors could include a short paragraph on how this issue could be handled with their software.

4) Chapter 3.4 Sea level correction: The first paragraph has some repetition regarding the short timescale, glacial/interglacial cycle driven sea-level changes. Should be partly rewritten.

5) Chapter 4: The description of input data formats for 2D and 3D is not always very clear. For instance, is with 'regular array' a constant spacing of nodes meant? And can the x,y coordinates be different from horizon to horizon, or must x,y be the same for all horizons? Please clarify.

6) Chapter 6: I think it would be very useful to potential users of the software if the authors could make the input files of their two cases accessible, either as Supplementary Material or by providing URLs. Most new users to a software prefer to start with well tested working example files to become more familiar with the parameterization before preparing and

loading their own data files. This would add to the acceptance and encouragement for a wider usage of PALEOSTRIP.

**Corrections of phrasing and spelling, and other minor issues:**

7: 'to allow users to'

12: '(e.g. the Shared Socio-economical Pathways by Riahi et al, 2017)'

15: '…, although …'

37: '…, and thus the paleo-water depth …'

45: 'trends'

50-51: '… of the geodynamical open-source software GPlates (https://www.gplates.org/). It benefits from geodynamical corrections related to kinematic, …'

55: '…areas to constrain climate …'

71: 'operating system'

81: 'operating system' and 'provide the PALEOSTRIP code'

84: 'Coordinate system'

101: I suggest to add '… on the thermal subsidence as well as erosion …'

Equation (1): Shouldn't it be $\rho_m + \rho_w$ in the last term?

111: Add abbreviation '… sea level (SL) variations …' to refer to term in equation.

123+125: 'right-hand side'

128: '… are explained following …'

187: '… grid point results are independent from each other.'

206: Remove the a before '2d and 3D …'

210: 'by means of'

211: Remove one of the two in.

212: 'also allow to use' (remove s)

222: 'the stretched lithosphere thins due to cooling'

247: 'larger time elapsed since'

258: Add 'On a long time scale, sea level …'

259: '… and on shorter time scales, due to …'

258-267: There are some repetitions in this paragraph. Please rewrite.

290: '… dynamic topography occurs at long wavelength …'

308: 'PALEOSTRIP'

324: It says 90% in Fig. 4a. Please be consistent.

330: Please clarify if $dx$ and $dy$ must be identical for all horizons polygons.

331: It is not clear what is meant with scattered data as opposed gridded data. Do you mean data on an irregular grid as opposed to data on a regularly spaced grid?

345: Add sentence: 'Depths are related to present regional sea level.'

349: Add '… is the age of horizons at the base of the layers and …'

368: 'The match between …'

377: Why are results better than with Flex-Decomp? Do you have a short explanation?

381: 'The match between …'

385: '… also extract 2D transects or 1D wells from 3D …'

386: 'we provide two case studies'

389: 'Both cases are taken …'

392: '… Ross Sea paleo-bathymetry …'

396-397: Shouldn't it be 'DSDP Site 273' on the western Ross Sea shelf?

414: '… depth of the mid-Miocene unconformity …'

419-424: According to Fig. 4 in De Santis et al. (1999), the unconformity RSU4 and other unconformities pinch out to the seafloor due to glacial erosion. How have the eroded sediments been taken into account? See also my comment above.

436: Add 'It can process paleobathymetries for …'

439: 'allow users to insert'

440: Remove a before 'minor work'

**Figures:**

Fig. 4a & caption: The text (line 324) says 80%. Please correct.

Fig. 4b & caption: It is not clear if the squared grid is absolutely required. How about a rectangular grid, e.g. in cases of long but narrow continental shelves?

Fig. 8 caption: Change to 'DSDP Site 273'.

Fig. 9 caption: What are the references for the RSU-4 and basement grids?

---

## Referee Comment (RC2)

**Review for Colleoni et al. (Geoscientific Model Development: gmd-2021-78):**
*PALEOSTRIPv1.0 - a user-friendly 3D backtracking software to reconstruct paleo-bathymetries*

The authors present a MATLAB open-source software called PALEOSTRIPv1.0, which performs 1D, 2D and 3D backtracking of paleobathymetries. The robust and comparable calculation of paleobathymetries is an important factor in past climate reconstructions and providing the community with a widely accessible, comprehensive software package will prove incredible valuable step towards more detailed paleo-reconstructions.
For this review, I assessed the provided manuscript as well as the software package and manual attached to the software.
Overall, the manuscript is well written and gives a very detailed description of the process of calculating paleobathymetries. The example of the Ross Sea Embayment is clear and showcases the advantages of PALEOSTRIP clearly. Nevertheless, I like to advise the authors to amend some more information and resources for the profit of the users just starting working on paleobathymetric reconstructions, to avoid misconceptions and confusion. I detail these and some other comments below. Additionally, the text has some minor spelling and grammatical issues (especially when referring to software names), which should be rectified.

*I recommend to accept the manuscript after minor revisions*.

**General comments on the manuscript:**

1. *Backstripping vs. Backtracking:* Throughout the manuscript, the authors reiterate that there is a difference between backstripping and backtracking without clearly defining and distinguishing both at the beginning of the manuscript. A paragraph laying this groundwork as well as explaining to the reader the different usage is warranted. A constant mentioning of these similar concepts confuses the reader. I suggest, the authors define the process they are calculating with PALEOSTRIP at the beginning of the manuscript and use only one to the terms in the remainder of the manuscript, to avoid confusion.

2. *What can PALEOSTRIP provide & what can't it provide?*
   For any paleobathymetric reconstruction, it is important to clearly define which geological environments can be reconstructed with a given method. The authors mention various caveats, that every geological reconstruction holds, throughout the manuscript, but a central paragraph briefly summarizing limitations and advantages of PALEOSTRIP for the reader can help convey various matters, that the users need to be aware of. More technical aspects might also be a good addition to the manual:
   - Crustal Structure: Can PALEOSTRIP be used in areas underlain by oceanic crust or transitional crustal types, which might be common on wider continental margins? By including multiple $\beta$-factors and thermal subsidence various settings for the continental crust are available, which is great to reconstruct more complicated geological settings, but does this also translate to oceanic crust?
   - Dynamic Topography: Since PALEOSTRIP does not include a platekinematic component, the user needs to be careful to use appropriate dynamic topography models for the reconstruction time frame, since they are often related to a specific platekinematic model.
   - Lithological parameters: Could you point the readers to resources, where they might find general values to compare their lithological parameters (especially the decompaction coefficient) to, or use in case the area they are working in has not been cored?
   - Erosion and re-sedimentation: One crucial and difficult to quantify aspect of paleobathymetric reconstruction, especially in areas of polar continental shelves such as the example from the Ross Sea given here, is the removal of sediments by

advancing glaciers and potentially various stages of sedimentation prior to the current position. How does PALEOSTRIP deal with scenarios of multiple sedimentation cycles and how do the authors recommend users do account/correct/anticipate this process?

3. *Communication of new releases:* The authors mention various times throughout the manuscript as well as within the software manual that certain functionalities are likely to be developed in the future. How will the authors ensure, that they inform the community about new developments by themselves or other groups?

**Specific comments on the manuscript:**

*Abstract:* The abstract should include a brief explanation, why paleobathymetries are important and how they connect to paleomodels.

*Introduction:* L24 tectonic setting might not encompass all the different factors listed here. To underline the importance of paleobathymetric reconstructions, it would be useful to add brief examples on the processes and give the reader a window of how much bathymetry can actually change within certain timeframes.

L32/33 "overprinted information" what do the authors mean by this? Erosion and re-deposition?

L40 and following: This is a great overview on what is currently available for backstripping/tracking. This paragraph would be a great opportunity to emphasize, where the current gaps are, that PALEOSTRIP is closing.

*Model framework and requirements:*

I tested download and installation of the software package and am happy to report, that there were no issues on MATLAB2020. The interface is well done and easy to navigate. The example data worked and I could re-calculate the steps presented in the manuscript.

L73: incompatibility with GUIDE → this information might be more suitable for the manual.

Coordinate Systems: The resources mentioned here should also be added to the manual.

Input files: mention the examples attached to the software package.

*PALEOSTRIP: Backtracking:*

This entire paragraph is a detailed and very clear overview on the different steps of paleobathymetric reconstruction and various consequences to the seafloor development. Several caveats the user needs to account for are mentioned throughout the text. As already detailed above, I would recommend a summary paragraph explaining the advantages and limitations of PALEOSTRIP to be included here.

*PALEOSTRIP Grid Interpolation:*

The various details described here are very convoluted. This paragraph might profit from a careful re-write. During the final editing, Fig. 4 needs to be as close as possible to the describing text to convey the information needed.

*PALEOSTRIP validation:*

Although the reader can quickly refer to DeSantis et al. 1999, it would be practical, to reiterate how the used lithological parameters have been measured on DSDP 273 (logging along core, discreet samples...).

L369 and following: The authors compare PALEOSTRIP and Flex-Decomp pointing out a good fit and some understandable discrepancies. What would be considered a good fit and what kind of error margins can be assumed for this kind of reconstruction? A brief overview on potential error sources and error margins

should be added to aid with maximum/minimum scenarios and raise awareness in the user community.

*Case study: example of the Ross Sea*

The example of the Ross Sea illustrates the different processes of the software package and does not draw on specialist knowledge of the region. Although, this manuscript is not designed to interpret the paleobathymetry of the Ross Sea, a little bit more detail to the results might be useful to fully see the physical settings at play. In my opinion, this does not necessarily require a lot of additional text, but can be conveyed with some changes to the attached figures (see comments to figures below).

**Figures:**

*General comment*:

Some of the figures use red and green, which might create inaccessibility issues for visually impaired readers.

*Fig. 2:* With a clear definition of backtracking vs. backstacking, the caption can be decluttered.

*Fig. 3*: Clear conceptual figure! I suggest to add this diagram to the manual as well.

*Fig. 4:* This figure immensely helped with my understanding of the paragraph on *PALEOSTRIP Grid Interpolation.* In the final edit, this figure needs to be set as closely as possible to this crucial paragraph.

*Fig. 5:* Colour scale and km-scale are difficult to see on the blue background. Users unfamiliar with the Ross Sea might profit from indicating the basement highs, which are emerging in the reconstruction.

*Fig. 6:* Axis description on panel A differs from other panels and should be homogenized for better comparability.

*Fig. 9/10/11*: Please change the used colour scale to be able to differentiate between both ends (currently both high and low end of the scale saturate in the red). Given, that the text states, that a certain portion of the embayment becomes subaerial, this should easily be visible in Fig. 10 & 11 (maybe use same colour scale as in Fig. 5?)

---

## Author Comment (AC1)

**Answers to reviewers's comments**

**Reviewer 1:**

**Review of paper by Colleoni et al.: *PALEOSTRIPv1.0 – a user-friendly 3D backtracking software to reconstruct paleo-bathymetries;* submitted to GMD (MS No. gmd-2021-78)**

**General comments:**

This paper presents the rationale, methodology, physical/geophysical background and parameterization of new open-source software to calculate paleo-water depths in a 1D, 2D and 3D backtracking/backstripping procedure. The development of a relatively user-friendly open-source software for paleobathymetry is timely as the relevance for improved paleo-water depths for different time slices of the past geological history of ocean basins and continental margins is largely increased for urgently needed improved paleo-ice sheet and paleo-climate modelling results. Previously available software solutions have been lacking various aspects, such as full 3D flexural response, implementation of mantle-driven dynamic topography, sensitivity testing, open-access/open-source, and user-friendliness. All aspects are absolutely necessary to a research community-wide acceptance and usage of this software. The paper is well structured and written, and it contains the necessary and informative tables and figures. The methods and mathematical background are absolutely sound, and previously developed procedures are correctly cited. I recommend the paper to be accepted for publication after the following minor issues are addressed.

**Specific comments and suggestions** (numbers refer to line numbers of manuscript):

1) In general: Difference between backtracking and backstripping? I think that this is rather a semantic difference. In both approaches, the same fundamental principles and equation systems are used. It's only a matter which of the variables should be solved or kept constant first: Either assuming a fixed paleo-water depth and then solve for decompaction and subsidence, or implying an assumed subsidence and decompaction for obtaining a paleo-water depth. I suggest to generally use the term backstripping for either approach as this term is widely known and recognised in the geoscience community.

*Many softwares in literature distinguish between backstripping and backtracking. Thus it is just a matter of terminology. Since we explicitly describe the differences within the manuscript between backtracking and backstripping, and also show the differences in the equations, we think it is more honest to distinguish between them. In PALEOSTRIP, it is not straightforward to perform a classical backstripping procedure since one cannot prescribe a paleo water depth history in this version of the software. The only way of doing backstripping is to deactivate the tectonic subsidence from the GUI and prescribe a water depth in the sea level tab of the GUI. In this version, the sea level tab does not allow to have a time-evolving*

*and spatially varying sea level history (only timeseries can be prescribed to have time-evolving values). With a few modifications, this can be done quite easily. But right now, the software is more dedicated to reconstruct paleo-water depths. Thus, we prefer keeping the terminology as it is for now and follow modern works terminology for basin analysis (e.g. Muller et al., 2018).*

3) Introduction: I suggest to add the following two references as previous examples of 3D backstripping cases from the SW African and Brazilian margins to the Introduction: Dressel et al. (Marine & Petroleum Geology, 2015) and Dressel et al. (Tectonophysics, 2017).

*Thank you very much for pointing out those references. We added them to the indicated paragraph.*

2) Chapter 3: The authors describe with great clarity all necessary steps for backtracking/backstripping (decompaction, isostatic correction, thermal subsidence, sea-level, dynamic topography), but one important aspect is missing which is in particular important for polar continental shelves. That is the reconstruction of sediments that were eroded/truncated by grounded ice-sheet advances. Of course, reconstructing such eroded sediments above truncational unconformities or the seafloor (in case where dipping sequences pinch out) requires assumptions on pre-erosional sedimentation rates, but it would be useful if the authors could include a short paragraph on how this issue could be handled with their software.

*This is a very good point. PALEOSTRIP cannot handle erosion at this stage. This would imply a varying number of sedimentary layers during computation, which is not implemented in this version. As you mentioned, one needs to make some initial hypothesis about how much and from where sediments are eroded and add them artificially as a single additional layer during backward modeling. A very good description of the procedure is provided in "Physical principles of sedimentary basin analysis" by Magnus Wangen (2010), chapter 5, 5.4 Erosion. This could be a way of dealing with it while modifying PALEOSTRIP structure. We added this short explanation to the manuscript (a new subsection 3.6).*

4) Chapter 3.4 Sea level correction: The first paragraph has some repetition regarding the short timescale, glacial/interglacial cycle driven sea-level changes. Should be partly rewritten.

*We simplified this paragraph and reformulated it. See the track-changed version of this manuscript.*

5) Chapter 4: The description of input data formats for 2D and 3D is not always very clear. For instance, is with 'regular array' a constant spacing of nodes meant? And can the x,y coordinates be different from horizon to horizon, or must x,y be the same for all horizons? Please clarify.

*Thanks for pointing this out, it was indeed misleading. We reformulated the sentences. Coordinates must be strictly identical for all horizons. Input data must be on the same grid. Spacing between each x and y must be constant. For 3D grids, the spacing along the X*

direction can differ from the spacing in the Y direction. But along the X direction, spacing must be constant. Same along the Y direction.

6) Chapter 6: I think it would be very useful to potential users of the software if the authors could make the input files of their two cases accessible, either as Supplementary Material or by providing URLs. Most new users to a software prefer to start with well tested working example files to become more familiar with the parameterization before preparing and loading their own data files. This would add to the acceptance and encouragement for a wider usage of PALEOSTRIP.

*All input files of case studies shown in this article are distributed on GitHub alongside with PALEOSTRIP code. There is a specific zip file named "paleostrip_examples". Actually, Reviewer 2 found them and used them to test PALEOSTRIP without any issues.*

**Corrections of phrasing and spelling, and other minor issues:**

7: 'to allow users to'

*Done*

12: '(e.g. the Shared Socio-economical Pathways by Riahi et al, 2017)'

*Done*

15: '..., although ...'

*Done*

37: '..., and thus the paleo-water depth ...'

*Done*

45: 'trends'

*Done*

50-51: '... of the geodynamical open-source software GPlates (https://www.gplates.org/). It benefits from geodynamical corrections related to kinematic, ...'

*Done*

55: '...areas to constrain climate ...'

*We actually chose a different formulation: "...areas that will be prescribed as boundary conditions within climate and ice sheet models". This is more correct than the previous formulation or the one you suggest here.*

71: 'operating system'

*Done*

81: 'operating system' and 'provide the PALEOSTRIP code'

*Done*

84: 'Coordinate system'

*Done*

101: I suggest to add '... on the thermal subsidence as well as erosion ...'

*Agreed and done.*

Equation (1): Shouldn't it be $\rho_m+\rho_w$ in the last term?

*No, this term is correct. This $\rho_m-\rho_w$ represents mass substitution, here water by water.*

111: Add abbreviation '... sea level (SL) variations ...' to refer to term in equation.

*Agreed and done.*

123+125: 'right-hand side'

*Thank you for pointing this out! Done.*

128: '... are explained following ...'

*Done*

187: '... grid point results are independent from each other.'

*Done*

206: Remove the a before '2d and 3D ...'

*Done*

210: 'by means of'

*Done*

211: Remove one of the two in.

*Done*

212: 'also allow to use' (remove s)

*Done*

222: 'the stretched lithosphere thins due to cooling'

*We reformulated with "the stretched lithosphere constricts due to cooling"*

247: 'larger time elapsed since'

*Done*

258: Add 'On a long time scale, sea level ...'

*Done*

259: '... and on shorter time scales, due to ...'

*Done*

258-267: There are some repetitions in this paragraph. Please rewrite.

*We did simplify this paragraph. See the track changes in the manuscript.*

290: '... dynamic topography occurs at long wavelength ...'

*Done*

308: 'PALEOSTRIP'

*Done*

324: It says 90% in Fig. 4a. Please be consistent.

*We corrected it. Correct value is 90%. Thanks for pointing this out!*

330: Please clarify if *dx* and *dy* must be identical for all horizons polygons.

*We clarified the introductory paragraph of Section 4 (see our answer to your previous general comment at the beginning of this review). And we further simplified the other paragraphs. We hope this is more readable in the new revised text.*

331: It is not clear what is meant with scattered data as opposed gridded data. Do you mean data on an irregular grid as opposed to data on a regularly spaced grid?

*Scattered data means that points are unstructured and are not written in a file following a classical NX*NY structure but treated as independent single points. When data is treated as scattered, this takes more computational time, but this also allows one to input either structured gridded data, or irregular polygon data (that can't really be read as a rectangular grid because some of the grid points would be empty).*

345: Add sentence: 'Depths are related to present regional sea level.'

*Done*

349: Add '... is the age of horizons at the base of the layers and ...'

*Done*

368: 'The match between ...'

*Done*

377: Why are results better than with Flex-Decomp? Do you have a short explanation?

*Yes. This is because TAFI is an isostasy scheme and we implemented it within PALEOSTRIP to test different ways of computing isostasy. However, the regridding of the loads is done by PALEOSTRIP in both cases. This explains why PALEOSTRIP finite difference isostasy and TAFI isostasy show only minor discrepancies. Flex-Decomp applies a different reggriding method of the loads, which mostly explains the discrepancies observed with PALEOSTRIP.*

*We added "Re-interpolation of the load is performed by PALEOSTRIP in both cases" to the description of TAFI impacts on backstripping.*

381: 'The match between ...'

*Done*

385: '... also extract 2D transects or 1D wells from 3D ...'

*Done*

386: 'we provide two case studies'

*Done*

389: 'Both cases are taken ...'

*Done*

392: '... Ross Sea paleo-bathymetry ...'

*Done*

396-397: Shouldn't it be 'DSDP Site 273' on the western Ross Sea shelf?

*Many thanks!...initially we used DSDP site 270 to provide a case study but we ultimately decided on site DSDP 273. We corrected it here and in the caption of Figure 8.*

414: '... depth of the mid-Miocene unconformity ...'

*Done*

419-424: According to Fig. 4 in De Santis et al. (1999), the unconformity RSU4 and other unconformities pinch out to the seafloor due to glacial erosion. How have the eroded sediments been taken into account? See also my comment above.

*See our answer to your general comment above about erosion treatment within PALEOSTRIP.*

436: Add 'It can process paleobathymetries for ...'

*Done*

439: 'allow users to insert'

*Done*

440: Remove a before 'minor work'

*Done*

**Figures:**

Fig. 4a & caption: The text (line 324) says 80%. Please correct.

*Thanks for pointing this out! We corrected the main text. The Figure's caption is correct.*

Fig. 4b & caption: It is not clear if the squared grid is absolutely required. How about a rectangular grid, e.g. in cases of long but narrow continental shelves?

*We apologize...we realise that the figure can be misleading. Actually, the grid is rectangular in the sense that PALEOSTRIP builds it based on Xmin and Xmax and Ymin and Ymax. So in the case of an elongated area, the corresponding grid will be rectangular and elongated in the direction of the largest width.*

Fig. 8 caption: Change to 'DSDP Site 273'.

*Many thanks, we corrected it.*

Fig. 9 caption: What are the references for the RSU-4 and basement grids?

*References are from the ANTOSTRAT project. We added them to the caption and to the main text (Subsection 6.2).*

---

## Author Comment (AC2)

**Answers to reviewers's comments**

**Reviewer 2:**

**Review for Colleoni et al. (Geoscientific Model Development: gmd-2021-78):**

*PALEOSTRIPv1.0 - a user-friendly 3D backtracking software to reconstruct paleo-bathymetries*

The authors present a MATLAB open-source software called PALEOSTRIPv1.0, which performs 1D, 2D and 3D backtracking of paleobathymetries. The robust and comparable calculation of paleobathymetries is an important factor in past climate reconstructions and providing the community with a widely accessible, comprehensive software package will prove incredible valuable step towards more detailed paleo-reconstructions.

For this review, I assessed the provided manuscript as well as the software package and manual attached to the software.

Overall, the manuscript is well written and gives a very detailed description of the process of calculating paleobathymetries. The example of the Ross Sea Embayment is clear and showcases the advantages of PALEOSTRIP clearly. Nevertheless, I like to advise the authors to amend some more information and resources for the profit of the users just starting working on paleobathymetric reconstructions, to avoid misconceptions and confusion. I detail these and some other comments below. Additionally, the text has some minor spelling and grammatical issues (especially when referring to software names), which should be rectified.

*I recommend to accept the manuscript after minor revisions.*

**General comments on the manuscript:**

1. *Backstripping vs. Backtracking: Throughout the manuscript, the authors reiterate that there is a difference between backstripping and backtracking without clearly defining and distinguishing both at the beginning of the manuscript. A paragraph laying this groundwork as well as explaining to the reader the different usage is warranted. A constant mentioning of these similar concepts confuses the reader. I suggest, the authors define the process they are calculating with PALEOSTRIP at the beginning of the manuscript and use only one to the terms in the remainder of the manuscript, to avoid confusion.*

*We disagree with the reviewer's comment since the entire Section 3 is dedicated to explain, by means of equations and text (and even in the section title) that PALEOSTRIP performs backtracking. Reviewer 1 stated that he understood the difference well and also re-explained it very well in its own comments. We were very careful of using "backtracking" or "backtracked" throughout the manuscript to be consistent. Thus we estimate that no further clarification is needed here.*

2. *What can PALEOSTRIP provide & what can't it provide?*

*For any paleobathymetric reconstruction, it is important to clearly define which geological environments can be reconstructed with a given method. The authors mention various caveats, that every geological reconstruction holds, throughout the manuscript, but a central paragraph briefly summarizing limitations and advantages of PALEOSTRIP for the reader can help convey various matters, that the users need to be aware of. More technical aspects might also be a good addition to the manual:*

- *Crustal Structure: Can PALEOSTRIP be used in areas underlain by oceanic crust or transitional crustal types, which might be common on wider continental margins? By including multiple β-factors and thermal subsidence various settings for the continental crust are available, which is great to reconstruct more complicated geological settings, but does this also translate to oceanic crust?*

*We agree on this point and we inserted a new sub-section 3 in which we added a sentence mentioning that PALEOSTRIP does not backtrack sediments on oceanic crusts for now. This could be the object of a future release with the help of interested users.*

- *Dynamic Topography: Since PALEOSTRIP does not include a platekinematic component, the user needs to be careful to use appropriate dynamic topography models for the reconstruction time frame, since they are often related to a specific platekinematic model.*

*Well, this recommandation holds for all input data. The present article is not focused on dynamic topography and we assume that the users are careful enough to choose appropriate input data for their reconstruction. Actually, as clearly stated in subsection 3.5, PALEOSTRIP does not provide any dynamic topography data. We just provided some regridded onto the Ross Sea sector for the sake of the case studies of this paper:"Note that Muller et al. 2018a is not implemented within PALEOSTRIP" .*

*We nevertheless added a sentence in subsection 3.5: "Maps of dynamic topography are inputs to PALEOSTRIP and the user is free to use any reconstructions. Note that inputs of dynamic topography require some post-processing to be adjusted to the area of interest before being passed through the GUI".*

- *Lithological parameters: Could you point the readers to resources, where they might find general values to compare their lithological parameters (especially the decompaction coefficient) to, or use in case the area they are working in has not been cored?*

*In other commercial softwares, standard lithological parameters are implemented already within the softwares. However the approach of PALEOSTRIP is to provide the physical structure and let the users provide all input data related to lithological parameters. Most of the books about basin analysis (Allen and Allen, 2013, Wangen et al., 2010, and many others) or some key papers such as Kominz et*

*al., (2011) provide analysis and explanation on how to retrieve those parameters. So by doing a bit of bibliography, one might retrieve the necessary parameters. The use of lithological parameters in areas for which lithology is unknown is at the discretion of the users and might vary from area to area.*

- *Erosion and re-sedimentation: One crucial and difficult to quantify aspect of paleobathymetric reconstruction, especially in areas of polar continental shelves such as the example from the Ross Sea given here, is the removal of sediments by advancing glaciers and potentially various stages of sedimentation prior to the current position. How does PALEOSTRIP deal with scenarios of multiple sedimentation cycles and how do the authors recommend users do account/correct/anticipate this process?*

*This is a very good point. PALEOSTRIP cannot handle erosion at this stage. This would imply a varying number of sedimentary layers during computation, which is not implemented in this version. How to deal with it? One needs to make some initial hypothesis about how much and from where sediments are eroded and add them artificially as a single additional layer during backward modeling. A very good description of the procedure is provided in "Physical principles of sedimentary basin analysis" by Magnus Wangen (2010), chapter 5, 5.4 Erosion. This could be a way of dealing with it while modifying PALEOSTRIP structure. We added this short explanation to the manuscript (a new subsection 3.6).*

3. *Communication of new releases:* The authors mention various times throughout the manuscript as well as within the software manual that certain functionalities are likely to be developed in the future. How will the authors ensure, that they inform the community about new developments by themselves or other groups?

*The updates will be the object of peer-reviewed articles in literature. Most of the functionalities that you are referring to will likely be developed within the next two years because this paper is funded by a national Italian project that will end in May 2023. However, because of the open source license, the philosophy is to let users free to implement their own developments within PALEOSTRIP and communicate them to the community or not. If there will be some interest to build a PALEOSTRIP community, many of those individual developments could be incoporated within PALEOSTRIP next releases. For now, those who are interested to follow some updates by our group can contact any of us directly.*

**Specific comments on the manuscript:**

*Abstract: Introduction:*

The abstract should include a brief explanation, why paleobathymetries are important and how they connect to paleomodels.

*This paper is really focusing on the description of the software. Thus we do not think it is important to add this sentence to the abstract. We prefer keeping it focused. The importance of paleo-bathymetric reconstructions is touched upon in the Introduction.*

L24 tectonic setting might not encompass all the different factors listed here. To underline the importance of paleobathymetric reconstructions, it would be useful to add brief examples on the processes and give the reader a window of how much bathymetry can actually change within certain timeframes.

*We added some references at the end of this sentence: "(e.g. Herold et al., 2008; Frigola et al., 2018; Müller et al., 2018a; Straume et al., 2020; Hochmuth et al., 2020)" to provide references for the readers about global paleogeography changes through different time periods.*

L32/33 "overprinted information" what do the authors mean by this? Erosion and re-deposition?

*We reformulated: "eroded and/or reworked", which is definitely more correct here. Thanks for pointing this out!*

L40 and following: This is a great overview on what is currently available for backstripping/tracking. This paragraph would be a great opportunity to emphasize, where the current gaps are, that PALEOSTRIP is closing.

*PALEOSTRIP's original contribution is mentioned in the next two paragraphs just after this one. We just think that no additional sentences are needed.*

*Model framework and requirements:*

I tested download and installation of the software package and am happy to report, that there were no issues on MATLAB2020. The interface is well done and easy to navigate. The example data worked and I could re-calculate the steps presented in the manuscript.

L73: incompatibility with GUIDE this information might be more suitable for the manual.

*We retain that it is important to have this info here so that at first glance, the reader can understand or not if his/hers operating system and software release is compatible with PALEOSTRIP without having to download the code and go through the Manual.*

Coordinate Systems: The resources mentioned here should also be added to the manual.

*Same here, this paper describes the software, this is the aim of such a journal. So all info useful to run the software should be mentioned here.*

Input files: mention the examples attached to the software package.

*Agreed. We added the following sentence: In the present study, the input data files associated to each case studies are zipped in paleostrip_examples.zip, available on GitHub at https://github.com/flocolleoni/PALEOSTRIPv1.0.*

*PALEOSTRIP: Backtracking:*

This entire paragraph is a detailed and very clear overview on the different steps of paleobathymetric reconstruction and various consequences to the seafloor development. Several caveats the user needs to account for are mentioned throughout the text. As already detailed above, I would recommend a summary paragraph explaining the advantages and limitations of PALEOSTRIP to be included here.

*As you reported, most of the caveats of the different aspects of PALEOSTRIP are discussed through the text. However, as you suggested, we inserted a new subsection 3.6 "Sediment erosion" and we inserted a sentence about applicability for oceanic crust at the end of the introductory paragraph of section 3: Note that the physics implemented does not allow the treatment of oceanic crust in this version. This can be done by adding a few more options in the GUI mainly for thermal subsidence (e.g., Muller et al., 2018b).*

*PALEOSTRIP Grid Interpolation:*

The various details described here are very convoluted. This paragraph might profit from a careful re-write. During the final editing, Fig. 4 needs to be as close as possible to the describing text to convey the information needed.

*Reviewer 1 also pointed out some confusing statement. We simplified it and we hope that the new version of this section is clearer.*

*PALEOSTRIP validation:*

Although the reader can quickly refer to DeSantis et al. 1999, it would be practical, to reiterate how the used lithological parameters have been measured on DSDP 273 (logging along core, discreet samples...).

*This paper is not a sedimentological description of the cores, but a description of the functionalities of a software. We thus think that this is not useful to the aim of this specific paper.*

L369 and following: The authors compare PALEOSTRIP and Flex-Decomp pointing out a good fit and some understandable discrepancies. What would be considered a good fit and what kind of error margins can be assumed for this kind of reconstruction? A brief overview on potential error sources and error margins should be added to aid with maximum/minimum scenarios and raise awareness in the user community.

*Uncertainties on edges influence on flexural isostasy is very well known (e.g. Wickert et al., 2016 provide some description on how to treat the load at the edge of a flexural grid) but only sediment cores can constran paleo-altitude or paleo-depths. Thus good reconstructions should result from the tuning of your flexural parameters (and of the other processes constrained by sediment core records. If the user doesn't have any observations to constrain*

*his/hers backtracked depths, then a statistical approach, performing ensembles of reconstructions varying the parameters within defined ranges (e.g., Monte Carlo, Latin Hypercube) is the only way of calculating statistical uncertainty and providing a measure of how good your backtracking is.*

*Potentially all the processes that are parameterized within PALEOSTRIP are a source of uncertainties, even the input paleodata by the users. There is no way to provide ad-hoc ranges of uncertainties for those processes since it will depend on the input data and on the area considered for backtracking.*

*Case study: example of the Ross Sea*

**Figures:**

The example of the Ross Sea illustrates the different processes of the software package and does not draw on specialist knowledge of the region. Although, this manuscript is not designed to interpret the paleobathymetry of the Ross Sea, a little bit more detail to the results might be useful to fully see the physical settings at play. In my opinion, this does not necessarily require a lot of additional text, but can be conveyed with some changes to the attached figures (see comments to figures below)

*As you point out this paper is not focusing on Ross Sea paleo-bathymetric reconstructions. Thus we did not modify the main text. But we improved the clarity of the captions of the figures to allow a better understanding.*

*General comment*:

Some of the figures use red and green, which might create inaccessibility issues for visually impaired readers.

*We acknowledge your suggestion. Those figures are direct PALEOSTRIP outputs and this colorscale is the PALEOSTRIP default colorscale. We wanted to illustrate the functionalities of the software. Contours are also provided on the Figures and we believe the readers might appreciate the differences even without colors.*

*Fig. 2:* With a clear definition of backtracking vs. backstacking, the caption can be decluttered.

*Agreed. We simplified the caption since it was clearly representing backtracking.*

*Fig. 3*: Clear conceptual figure! I suggest to add this diagram to the manual as well.

*Thanks!*

*Fig. 4:* This figure immensely helped with my understanding of the paragraph on *PALEOSTRIP Grid Interpolation.* In the final edit, this figure needs to be set as closely as possible to this crucial paragraph.

*Thanks!*

*Fig*. *5:* Colour scale and km-scale are difficult to see on the blue background. Users unfamiliar with the Ross Sea might profit from indicating the basement highs, which are emerging in the reconstruction.

*We put both scales in a white box to improve readability. Since we are not using any Ross Sea bathymetric highs in the main text, this is not relevant info to add to this figure.*

*Fig*. *6:* Axis description on panel A differs from other panels and should be homogenized for better comparability.

*Done.*

*Fig*. *9/10/11*: Please change the used colour scale to be able to differentiate between both ends (currently both high and low end of the scale saturate in the red). Given, that the text states, that a certain portion of the embayment becomes subaerial, this should easily be visible in Fig. 10 & 11 (maybe use same colour scale as in Fig. 5?).

*Actually, only the island in the uppermost right corner of the figure is above sea level. All the rest is below sea level. Those figures differ from Fig 5. They are PALEOSTRIP output figures and the colorscale is PALEOSTRIP default colarscale. We now indicate all this in the captions of Fig. 9, 10 and 11: "Layout is from PALEOSTRIP plotting GUI and the colorscale is the default colorscale impemented within PALEOSTRIP and has been saturated below -4000 meters and above 200 meters for the need of this figure. The island located on the uppermost right corner is the only location with elevation above sea level".*

---

## Author Response (AR2)

**Answers to Topical Editor comments**

*Dear Editor, we provide below the answers to your suggestions and comments, in grey italic font. We hope our answers will be satisfactory.*

Dear Dr. Colleoni and co-authors,

After two very positive reviews and your responses and changes, I am happy to recommend your work for publication in GMD, following minor revisions based around the comments that I leave below:

1. The name of the software. I note your "backtracking" vs. "backstripping" argument -- something that is new to me -- and happily defer to your expertise here. However, this naming seems a bit incongruent with the name of the software ("PALEOSTRIP"). I imagine that backstripping is a more well known procedure... perhaps this is the reason for the name? And might you consider changing it to better match what the software does?

*Has you said, PALEOSTRIP can also do backstripping (because the equation is the same) but the main interface is more backtracking oriented. So the name remains pertinent here. Also, it has already been downloaded many times and we got already people using it so we prefer not changing the name at this stage. This is also true that we are developing a new interface to choose in which direction to use the main equation: backtracking or backstripping.*

2. "Scattered data". You answered the referee's comment in the response, but I think that it would be better to also make the text a bit clearer in this regard.

*You are right. We added the following sentence within the manuscript directly, at line 347 : "This means that points are unstructured and are not written in a file following a classical NX\*NY structure but treated as independent single points. When data is treated as scattered, this takes more computational time, but this also allows one to input either structured gridded data, or irregular polygon data"*

3. "Note that inputs of dynamic topography require some post-processing to be adjusted to the area of interest before being passed through the GUI."
Is this just ensuring that the dynamic topography inputs are aligned and clipped to the model domain and cell size. If so, this seems trivial (so maybe no need to mention), but perhaps you could make it more clear (to me "some post processing" may imply vague additional changes that need to be made).

*Thanks for pointing this out! We meant "pre-processing"… So we corrected it within the manuscript (line 316).*

4. It could be helpful to include a table of suggested standard lithological parameters.

*We disagree on this point. This is because there are no such standard parameters. It is very difficult to retrieve the lithological properties for sediments and clays or silt or sand have a large spectrum of coefficients depending on the context in which they have been deposited and the sediment composition: sometimes it is not 100% sand, or clay or silt. Instead we inserted a reference Kominz et al., (2011) that explains how to retrieve them and also provides table of parameters in various IODP sites. This is a more honest approach. We added the following sentence at line : "Since the*

*lithological parameters varies a lot given the composition of sediments and their depositional context, we refer the reader to Kominz et al. (2011) for values and detailed explanations on how to retrieve the decompaction coefficients for the different lithologies of marine sediments."*

5. Both referees mention erosion and re-sedimentation. I understand that PALEOSTRIP does not at present include these. However, this seems a bit incompatible with your Ross Sea example. How would you propose to address this?

*No actually, this erosion and deposition issue is the same wherever you apply backstripping or backtracking on the globe. This is a real conceptual issue of paleo reconstructions. For example, in Paxman et al., (2019), they applied a correction for erosion a posteriori (after the backtracking). This is the only approach one can have and this is not an issue of PALEOSTRIP or the Ross Sea. The only thing we can do within PALEOSTRIP, in a future version, is to allow one to provide this correction during runtime, thus to account for related sediment weight at a different place. But this require to change the entire code structure and have flexible number of layers (new layers that at present have been eroded, but were there at a given time in the past).*

6. You should remove the mentions of new releases: We cannot see the future.

*Ok. We removed them.*

7. I might suggest including a bit about the definition and importance of backtracking to recover paleobathymetries in the abstract/introduction. This might be very helpful towards engaging readers who are interested in the problems that you address but who may be earlier in their careers / coming from different backgrounds, and who therefore do not yet know the vocabulary.

*Agreed. We added the following sentence in the abstract: "Reconstructing paleo-bathymetries is critical to better understand how oceanic circulation and ice sheets evolved through time and interacted with the different components of the Earth's system. Backtracking paleo-bathymetries implies reconstructions of these interactions in the past. Past reconstructions also directly serve as boundary conditions to numerical climate and ice sheet models, and as such, reliable reconstructions accounting for a maximum of sedimentary and solid Earth processes are necessary."*

*Since the interest of such procedure is already described in depth in the first and mostly the second paragraph of the introduction, we did not added this sentence to the introduction.*

8. I would suggest changing your GitHub repository naming/versioning structure. GitHub repositories are for a piece of code, whereas the releases (which can be integrated with Zenodo to provide automated doi tagging) are for the versions. Of course, I can't require this as a journal editor, but I would recommend it as good coding practice.

*Ok. Thank you for the advice. For the moment we will let it as it is. This is because we are developing a far much complex of it and it will be hosted in a different directory. There won't be branches to the current v1 of the code.*

9. Noting that a colorblind-unfriendly color scale is "the PALEOSTRIP default colorscale" is not a rebuttal, but rather an admission of a flaw in the accesibility of the software. Surely there should be a simple way to replace this wtih any number of other color scales? Perhaps this may help: https://www.fabiocrameri.ch/colourmaps/

*Actually the color scale can be changed easily on the MATLAB Figure interface with any other colorscale provided by MATLAB. We thus choose to let the colorscale as it is and instead insert a sentence in the figures 9, 10 and 11 caption to explain it.*

10. Diemsnions. Considering that 1D is a line and 2D is a plane, mathematically, I think that your 2D/3D flexural isostasy have an additional dimension added beyond those of the problem. (I do admit that van Wees and Cloetingh did the same…).

*Yes, that's always a bit tricky when explaining. It was correctly explained at lines 101-102. But we also improved slightly in section 4:*
- *For 2D, the title of the section was changed to "vertical transect" and the introductory sentence was inserted: "Vertical transects imply that input data are provided along an horizontal direction X and a vertical direction (depth) Z".*
- *For 3D, the following introductory sentence was inserted: "Maps implies that data are provided along the two horizontal directions X and Y and along the vertical direction Z".*

I look forward to seeing your revised manuscript.

*We hope our answers and corrections are satisfactory!*

---

## Author Response (AR3)

**Answers to Topical Editor comments**

*Dear Editor, we provide below the answers to your suggestions and comments, in grey italic font. We hope our answers will be satisfactory.*

**Topical Editor Decision: Publish subject to technical corrections** (12 Jul 2021) by Andrew Wickert
Comments to the Author:
Dear Dr. Colleoni and co-authors,

Thank you for your swift response and the associated changes and counterarguments. I have a few quick suggestions, but that is all.

For your abstract, I'd like to offer the following suggestion in the place of the three new sentences, in case this helps you to reach the point about your model sooner and provides the same motivating information to the reader:

Paleo-bathymetric reconstructions provide boundary conditions to numerical models of ice-sheet evolution and ocean circulation, critical to understanding their evolution through time.

If you would like, it is also quite common to then have a second sentence of the approximate form "Currently, the scientific community lacks a tool to do X (which is preventing scientific advances)." Then the third sentence can be, "In response to this need, we ..."

Or something like this!

*Thank you very much for your suggestions. We adopted them and the beginning of the abstract now looks like:*
*"Paleo-bathymetric reconstructions provide boundary conditions to numerical models of ice-sheet evolution and ocean circulation, critical to understanding their evolution through time. The geological community lacks of a complex open-source tool that allows for community implementations and can strengthen research synergies. To fill this gap, we present PALEOSTRIPv1.0…"*

The second point: I really do not like to push back against author rebuttals, but in this case I would really like to request a new color map for Fig. 9. The contour lines are think and unlabeled, and are therefore not sufficient for me to comprehend the plots if I turn off the color. And I do see this as an accessibility issue for our colleagues with disabilities. I am sorry that this will take you extra work -- I am sure that you are quite ready to see this paper be published!

https://www.nature.com/articles/s41467-020-19160-7

*We did a major implementation in the software: we implemented in total 12 color scales, 3 color scales are typical MATLAB ones (hsv, jet, parula), 9 color scales are taken from Crameri et al., (2020) and are color-blind friendly. Implications are:*
*- we needed to modify the GUI interface (and we revised Figure 7): so we committed the new version of the code on GitHub and substituted the release on Zenodo.*

- *We modified Figures 9, 10, 11 and A1, and we added the following sentences in the captions: "Note that the color scale is color-blind friendly and is from Crameri et al. (2020). PALEOSTRIP has implemented 12 different color scales, 9 out of 12 are color-blind friendly".*

- *We added the following sentence in the introductory paragraph of section 2: "Note that the final version related to this study has been corrected with minors bugs and the main ``Plot \& Save'' graphical interface has been modified to implement 12 color scales, among which, 8 color-blind friendly scales from Crameri et al., (2020). We thus invite the readers to download again the code from Github and from Zenodo."*

Beyond this, I think that your changes are satisfactory. I am suggesting "technical corrections" because of the minor nature of the requests, which will require no further review by me. Best wishes for the remaining steps leading to publication,

Andy

*Thank you very much for your support and your suggestions.*